

# Adaptive resilient containment control using reinforcement learning for nonlinear stochastic multi-agent systems under sensor faults

Guanzong Mo[1] and  Yixin Lyu[2]

[1] Guangdong University of Technology, Canton, China
[2] Xiamen University, Xiamen, China

## ABSTRACT

This article proposes an optimized backstepping control strategy designed for a category of nonlinear stochastic strict-feedback multi-agent systems (MASs) with sensor faults. The plan formulates optimized solutions for the respective subsystems by designing both virtual and actual controls, achieving overall optimization of the backstepping control. To address sensor faults, an adaptive neural network (NN) compensation control method is considered. The reinforcement learning (RL) framework based on neural network approximation is employed, deriving RL update rules from the negative gradient of a simple positive function correlated with the Hamilton-Jacobi-Bellman (HJB) equation. This significantly simplifies the RL algorithm while relaxing the constraints for known dynamics and persistent excitation. The theoretical analysis, based on stochastic Lyapunov theory, demonstrates the semi-global uniform ultimate boundedness (SGUUB) of all signals within the enclosed system, and illustrates the convergence of all follower outputs to the dynamic convex hull defined by the leaders. Ultimately, the proposed control strategy's effectiveness is validated through numerical simulations.

## INTRODUCTION

Multi-agent systems (MASs) have garnered considerable attention due to their ability to organize vast and intricate systems into smaller, intercommunicating, easily coordinated, and manageable subsystems. Currently, MASs find widespread applications in various domains such as aircraft formation, sensor networks, data fusion, parallel computing and cooperative control of multiple robots (*Antonio et al., 2021*; *Tang et al., 2016*; *Liu et al., 2020*; *Zhao et al., 2023*; *De Sá & Neto, 2023*). As a class of classical control problems from cooperative control, the containment control approach guarantees the convergence of all followers to a dynamic convex hull formed by multiple leaders. Numerous findings on containment control have been documented in the last decade (*Li et al., 2022*; *Li, Pan & Ma, 2022*; *Li et al., 2023*; *Liang et al., 2021*).

Corresponding author
Yixin Lyu, dukx15@163.com

It is noteworthy that optimal control, formally introduced by *Bellman (1957)* and *Pontryagin et al. (1962)* half a century ago, has become the foundation and prevailing design paradigm of modern control systems. The key to solving the optimal control problem lies in solving the Hamilton–Jacobi–Bellman (HJB) equation. Theoretically, solving optimal control based on the HJB equation is nearly impossible using analytical methods due to its strong nonlinearity (*Beard, Saridis & Wen, 1996*). Fortunately, Werbos (*Werbos, 1992*) introduced the approximate technique referred to as Adaptive Dynamic Programming (ADP) or Reinforcement Learning (RL), providing an effective method for solving the HJB equation. To date, this technique has witnessed significant development and achievements, as seen in *Wen, Xu & Li (2023)*, *Chen, Dai & Dong (2022)*, *Gao & Jiang (2018)*, *Zargarzadeh, Dierks & Jagannathan (2012)*, *Zargarzadeh, Dierks & Jagannathan (2012)*, *Li, Sun & Tong (2019)*, *Song & Dyke (2013)*, *Hu & Zhu (2015)*, *Rajagopal, Balakrishnan & Busemeyer (2017)*, *Wen, Xu & Li (2023)*. In *Wen, Xu & Li (2023)*, RL was combined with backstepping to design actual controls and virtual controls, optimizing the overall control of high-order systems. In *Chen, Dai & Dong (2022)*, this technique was applied to underactuated surface vessels, ensuring optimal tracking performance for ship control. *Gao & Jiang (2018)* addressed the computation problem of adaptive nearly optimal trackers without prior knowledge of system dynamics. In *Zargarzadeh, Dierks & Jagannathan (2012)*, investigated neural network-based adaptive optimal control for nonlinear continuous-time systems with known dynamics in strict-feedback form. *Zargarzadeh, Dierks & Jagannathan (2015)* extended their work to address nonlinear continuous-time systems characterized by uncertain dynamics in strict feedback form. They accomplished this by adapting the standard backstepping technique, as outlined in *Zargarzadeh, Dierks & Jagannathan (2015)*, transforming the optimal tracking problem into an equivalent optimal control problem and generating adaptive control inputs. *Li, Sun & Tong (2019)* presented a data-driven robust approximate optimal tracking scheme for a subset of strict-feedback single-input, single-output nonlinear systems characterized by the presence of unknown non-affine nonlinear faults and unmeasured states. In addition to deterministic nonlinear systems, various optimal control methods have been explored for stochastic systems in the past decade. The numerical techniques, proposed by *Song & Dyke (2013)*, aimed to reduce system responses under extreme loading conditions with stochastic excitations. *Hu & Zhu (2015)* introduced a stochastic optimization-based bounded control strategy for multi-degree-of-freedom strongly nonlinear systems. In *Rajagopal, Balakrishnan & Busemeyer (2017)*, an offline ADP method based on neural networks was developed to address finite-time stochastic optimal control problems. Specifically, in *Wen, Xu & Li (2023)* applied the RL strategy with the actor-critic architecture to stochastic nonlinear strict-feedback systems. However, for more complex nonlinear stochastic MASs, the above methods have not been fully studied. The challenges lie in the stability analysis process where the quadratic form of the Lyapunov function is no longer applicable, necessitating a reproof of system stability. Furthermore, in contrast to the single-agent stochastic strict-feedback system discussed in *Wen, Xu & Li (2023a)*, we consider complex multi-agent systems. Many practical multi-agent systems, especially in areas like intelligent transportation and smart grids, tackle complex large-scale problems that surpass the

capabilities of individual nonlinear systems. Therefore, research on nonlinear multi-agent systems is more meaningful.

Furthermore, in real-world scenarios, MASs comprise numerous actuators and sensors. Faults of some actuators or sensors can lead to the deviation from global control objectives. Therefore, investigating fault-tolerant control for MASs can enhance their safety and reliability. For instance, *Ding et al. (2018)* applied a region-based segmentation analysis to overcome caused by multiple sensor faults in strict-feedback systems. *Wang et al. (2018)* introduced a fault model to achieve fault-tolerant consensus for a multi-vehicle wireless network system with different actuator faults. *Cao et al. (2021)* fully considered consensus problems in MASs with sensor faults, utilizing neural networks not only for identifying unknown nonlinearities but also for designing adaptive compensatory controllers. Although there have been studies related to sensor faults, the conclusions from the above research cannot be directly applied to randomly occurring systems with statistical characteristics.

Inspired by the discussions above, This paper presents an enhanced backstepping control method tailored for a class of nonlinear stochastic strict-feedback MASs experiencing sensor faults. The primary contributions are summarized as follows:

(1) In this article, the optimal backstepping (OB) control method is extended to the nonlinear stochastic MASs with multiple leaders, which is more general than the consensus control results of MASs and can solve the optimal containment control problem.

(2) Suppressing sensor faults is important to enhance the system's safety and reliability. To tackle the challenge posed by sensor faults in stochastic MASs, consideration is given to an adaptive neural network (NN) compensation control method. This method is designed to alleviate the adverse effects of sensor faults on the MASs.

(3) The proposed adaptive control scheme successfully solves the problem of contained control with sensor faults, and the designed RL optimization method can optimize the control of unknown or uncertain stochastic dynamic systems.

## PRELIMINARIERS AND PROBLEM FORMULATION
### Graph theory

In the context of a group of N + M agents, the associated directed graph $\mathfrak{G}$ can be described by $\mathfrak{G} = (\mathfrak{V}, \mathfrak{E}, \Lambda)$, where $\mathfrak{V} = 1, 2, \ldots, N, \ldots, N + M$ constitutes a set of nodes, and $\mathfrak{E} = (j, i) \in \mathfrak{V} \times \mathfrak{V}$ represents a set of edges. The adjacency matrix is $\Lambda = [a_{ij}] \in \mathbb{R}^{(N+M) \times (N+M)}$, $(j, i) \in \mathfrak{E}$ implies that nodes $j$ and $i$ can share information with one another. $a_{ij}$ is defined as

$$a_{ij} = \begin{cases} 1, \text{if } (i, j) \in \mathcal{E} \\ 0, \text{if } (i, j) \notin \mathcal{E} \end{cases} \tag{1}$$

where the set of neighbors for a node $i$ is denoted by $\mathfrak{N}_i = j \in \mathfrak{V} : (j, i) \in \mathfrak{E}$. The Laplacian matrix $\mathbb{L} = [l_{ij}]_{(M+N) \times (M+N)} = \mathfrak{D} - \Lambda \in \mathbb{R}^{(N+M) \times (N+M)}$ is defined as

$$l_{ij} = \begin{cases} -a_{ij}, \text{if } i \neq j \\ \sum_{j \in \mathcal{N}_i} a_{ij}, \text{if } i = j \end{cases} \tag{2}$$

where $\mathfrak{D} = \{\text{diag} d_1, \ldots, d_N\}$ represents the degree matrix and $d_i = \sum_{j \in \mathfrak{N}_i} a_{ij}$. In this paper, the focus is on N + M agents, comprising N followers and M leaders, within a directed graph topology. It is assumed that each follower has at least one neighbor. Consequently, one can observe

$$\mathbb{L} = \begin{bmatrix} \mathbb{L}_1 & \mathbb{L}_2 \\ 0_{M \times N} & 0_{M \times M} \end{bmatrix} \tag{3}$$

where $\mathbb{L}_1 \in \mathbb{R}^{N \times N}, \mathbb{L}_2 \in \mathbb{R}^{N \times M}$.

Assumption 1: Each follower is connected to a minimum of one leader through a directed path, while leaders themselves lack neighboring nodes.

Lemma 1: According to Assumption 1, the matrix $\mathbb{L}_1$ issymmetric and positive definite, each element of $-\mathbb{L}_1^{-1}\mathbb{L}_2$ is nonnegative scalar, and all row sums of $-\mathbb{L}_1^{-1}\mathbb{L}_2$ equal to 1.

Assumption 2: (*Yoo, 2013*) The multiple leaders' outputs $y_{ld}, l \in (N+1, \ldots, N+M)$ and their derivatives $\dot{y}_{ld}, \ddot{y}_{ld}, \ldots, y_{ld}^{(n)}$ are bounded.

Lemma 2: (*Tong et al., 2011a*) Existing continuously differentiable function $V(t, x) \in \mathbb{R}^+$, it meets the conditions

$$v_1(\|x\|) \le V(t, x) \le v_2(\|x\|) \tag{4}$$

$$\mathcal{L}V(t, x) \le -aV(t, x) + c \tag{5}$$

where a > 0, c > 0 are constants, $v1(\cdot)$, $v2(\cdot)$ are K $\infty$ functions, the differential Eq. (9) has a singular, robust solution, and subsequent inequality is satisfied:

$$\mathbb{E}[V(t, x)] \le e^{-at}V(0, x(0)) + \frac{c}{a}. \tag{6}$$

Inequality Eq. (6) signifies that the solution $x(t)$ showcases SGUUB when considering expectations.

Lemma 3: (*Wang, Wang & Peng, 2015*) Defining $s_{*1} = [s_{11}, s_{21}, \ldots, s_{N1}]^T$, $y_i = [y_1, y_2, \ldots, y_N]^T$ we have $s_{*1} = \mathcal{L}_1 y_i + \mathcal{L}_2 y_{ld}$. Then the following inequality holds:

$$\|y_i + \mathbb{L}_1^{-1}\mathbb{L}_2 y_{ld}\| \le \|s_{*1}\| / \|\bar{\eta}(\mathbb{L}_1)\| \tag{7}$$

where $\|\bar{\eta}(\mathcal{L}_1)\|$ is the minimum singular value of $\mathbb{L}_1$.

Lemma 4 (Young's Inequality (*Tong et al., 2011*)): For all $x, y \in \mathbb{R}+$, the subsequent inequality is held:

$$xy \le \frac{1}{p}x^p + \frac{1}{q}y^q \tag{8}$$

where $p > 0, q < 0, 1/p + 1/q = 1$.

## Stochastic systems statement

Consider a group of nonlinear stochastic MASs described as follows:

$$\begin{cases} dx_{im} = [x_{im+1} + f_{im}(\bar{x}_{im})]dt + \psi_{im}(\bar{x}_{im})dw \\ dx_{in} = [u_i + f_{in}(\bar{x}_{in})]dt + \psi_{in}(\bar{x}_{in})dw \\ y_i = h(x_{i1}) \end{cases} \tag{9}$$

where $\overline{x}_{im} = [x_{i1}, \ldots, x_{im}]^T \in \mathbb{R}^m$ ($m = 1, \ldots, n-1$) represents the state vector. $u_i \in \mathbb{R}$ denotes the control input, $y_i \in \mathbb{R}$ is the system output. $h(x_{i1}) = k_i(t)x_{i1} + \rho_i(t)$ ,where $k_i(t)$ and $\rho_i(t)$ denote the parameters of sensor faults. $f_{im}(\cdot) \in \mathbb{R}^m$ and $\psi_{im}(\cdot) \in \mathbb{R}^m$ depict uncertain smooth functions. $w \in \mathbb{R}^r$ denotes the independent r-dimensional standard Brownian motion.

## Neural network approximation

It has been shown that a neural network (NN) can approximate any continuous function $F(x) : \mathbb{R}^n \to \mathbb{R}^m$ to a desired accuracy within a specified compact set $\Omega_x \subset \mathbb{R}^n$. The neural network approximation function can be represented as follows:

$$F_{\mathrm{NN}}(x) = W^T S(x) \tag{10}$$

where $W \in \mathbb{R}^{q \times m}$ is the weight matrix, $q$ is the quantity of neurons, $S(x) = [s_1(x), \ldots, s_q(x)]^T \in \mathbb{R}^q$ is the Gaussian basis function vector with $s_i(x) = \exp(-(x-v_i)^T (x-v_i)/\varphi_i^2) \in \mathbb{R}$, $v_i = [v_{i1}, \ldots, v_{\mathrm{in}}]^T \in \mathbb{R}^n$ represents the centers of receptive fields, and $\varphi_i$ is the width of the Gaussian function.

To fulfill Eq. (10), there must exist an ideal weight $W^*$, and the function $F(x)$ can be rewritten as

$$F(x) = W^{*T} S(x) + \varepsilon(x) \tag{11}$$

where $\varepsilon(x) \in \mathbb{R}^m$ is the approximation error required to meet $||\varepsilon(x)|| \leq \delta$ with $\delta$ *being* a positive constant.

The ideal weight matrix $W^*$ can be shown as

$$W^* = \arg\min_{W \in \mathbb{R}^{p \times m}} \left\{ \sup_{x \in \Omega_x} ||F(x) - WS(x)|| \right\}. \tag{12}$$

The Eq. (12) implies that the NN approximation error in Eq. (11) represents the minimum achievable deviation between $F(x)$ and $W^T S(x)$.

## Sensor faults

Within sensor fault model (*Bounemeur, Chemachema & Essounbouli, 2018*), the unspecified parameters adhere to $0 < \overline{k}_{i\min} \leq k_i(t) \leq 1$ and $-\overline{\rho}_i \leq \rho_i(t) \leq \overline{\rho}_i$, where $\overline{k}_{i\min} > 0$ represents the minimum sensor effectiveness, $-\overline{\rho}_i$, $\overline{\rho}_i$ are the lower bound and the upper bound respectively. The parameters of the sensor fault models can be summarized as below:

(a) If $k_i(t) = 1$ and $\rho_i(t)$ is a constant, the sensor exhibits bias fault.
(b) If $k_i(t) = 1$, $|\rho_i(t)| = \iota t, 0 < \iota \ll 1$, the sensor experiences a drift fault.
(c) If $k_i(t) = 1$, $|\rho_i(t)| < \overline{\rho}_i, \rho_i(t) \to 0$, this signifies that the sensor has incurred a loss of accuracy.
(d) If $0 < \overline{k}_{i\min} \leq k_i(t) \leq 1$, $\rho_i(t) = 0$, this suggests that the sensor has undergone a loss of effectiveness.

Denote $f_{si} = (k_i(t) - 1)x_{i1} + \rho_i(t)$. Then $y_i$ can be reformulated as $y_i = x_{i1} + f_{si}$. The derivative of $y_i$ can be rewritten as $\dot{y}_i = \dot{x}_{i1} + f_{psi}$, where $f_{psi} = \dot{f}_{si}$.

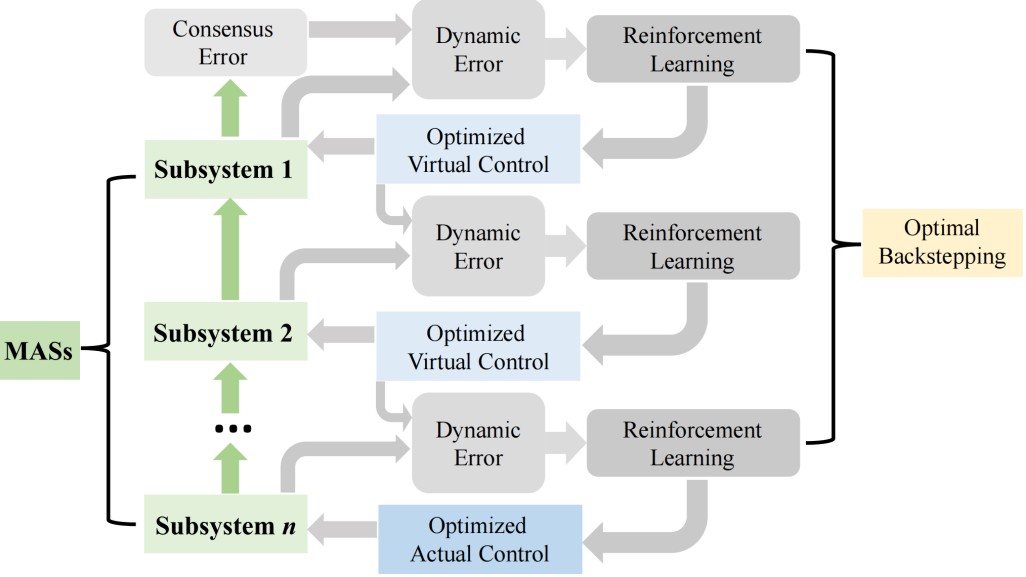

**Figure 1** OB design in the $i$ th agent, $i = 1, \ldots, n$.

## Operator $\mathfrak{L}$

For function $V(t, x)$, calculate its differential operator $\mathfrak{L}$ as *Mao, (2006)*

$$\mathcal{L}V = \frac{\partial V}{\partial x^T}(f(x) + g(x)u(x)) + \frac{1}{2}Tr\left\{\psi^T\frac{\partial^2 v}{\partial x \partial x^T}\psi\right\} \tag{13}$$

where Tr signifies the matrix trace.

## DISTRIBUTED ADAPTIVE OPTIMAL CONTAINMENT CONTROL

The backstepping technique is employed for controller design. Before we begin, to clearly demonstrate our ideas and process, let's provide a brief overview using Fig. 1.

Figure 1 illustrates the application process of RL in the design of optimized backstepping control. This process employs a Critic-Actor architecture to address the leader-following consensus control issue for nonlinear MASs. Within this method, the actor network is responsible for generating control actions, while the critic network evaluates the performance of the current control strategy. By iterating these two networks, the RL algorithm can learn an optimized control strategy that optimizes the control performance of the entire system.

Specifically, the optimal control problem is transformed into solving the HJB equation. However, due to the nonlinearity of the HJB equation, solving it directly is very challenging. To overcome this difficulty, a neural network-based RL method is proposed. This method derives the RL update rules from the negative gradient of a simple positive function, thereby avoiding the direct handling of multiple nonlinear terms in the HJB equation. This not only simplifies the algorithm but also relaxes the requirements for known system dynamics and persistent excitation.

During the RL learning process, the critic network first evaluates the performance of the current control strategy and provides it as feedback to the actor network. The actor network then adjusts its control actions based on this feedback, with the expectation of improving the system's performance. In this way, the RL algorithm can continuously learn and optimize the control strategy through iteration until the optimal solution is found.

To start with, the $i$-th subsystem's distributed containment error is defined as

$$s_{i1} = \sum_{j=1}^{N} a_{ij}(y_i - y_j) + \sum_{\ell=N+1}^{N+M} a_{i\ell}(y_i - y_{ld})$$

$$s_{im} = x_{im} - \alpha_{im-1}(m = 2, \ldots, n)$$
(14)

where $\alpha_{im-1}$ denotes the virtual controller. The OB control method is designed as follows.

Step 1: With Eq. (14) and Itô formula, the containment error can be calculated as follows:

$$ds_{i1} = \left[ d_i(x_{i2} + f_{psi} + f_{i1}(x_{i1})) - \sum_{j=1}^{N} a_{ij}(\dot{x}_{j1} + \dot{f}_{sj}) - \sum_{\ell=N+1}^{N+M} a_{il}\dot{y}_{ld} \right] dt +$$

$$\left[ d_i\psi_{i1}(x_{i1}) - \sum_{j=1}^{N} \psi_{j1}(x_{j1}) \right] dw = \left[ d_i x_{i2} - \sum_{j=1}^{N} a_{ij}x_{j2} + F_{i1} \right] dt + \Psi_{i1} dw$$

where:

$$F_{i1} = d_i(f_{psi} + f_{i1}(x_{i1})) - \sum_{j=1}^{N} a_{ij}(f_{psj} + f_{j1}(x_{j1})) - \sum_{\ell=N+1}^{N+M} a_{i\ell}\dot{y}_{\ell d}$$

$$\Psi_{i1} = d_i\psi_{i1}(x_{i1}) - \sum_{j=1}^{N} \psi_{j1}(x_{j1})$$

Representing virtual control by $\alpha_{i1}$, the performance index function is formulated as

$$J_{i1}(s_{i1}) = \int_{t}^{\infty} c_{i1}(s_{i1}(s), \alpha_{i1}(s_{i1}(s)))ds$$
(16)

where $c_{i1}(s_{i1}, \alpha_{i1}) = s_{i1}^2(t) + \alpha_{i1}^2$ is the cost function.

Replace $\alpha_{i1}$ with $\alpha_{i1}^*$ (optimal virtual control) in Eq. (16), the function is obtained as

$$J_{i1}^*(s_{i1}) = \int_{t}^{\infty} c_{i1}(s_{i1}(s), \alpha_{i1}^*(s_{i1}(s)))ds$$
(17)

According to the previous introduction, the function is given as follows:

$$\mathbb{E}[J_{i1}^*(s_{i1})] = \min_{\alpha_{i1} \in \Psi(\Omega)} \left[ \mathbb{E}[\int_{t}^{\infty} c_{i1}(s_{i1}, \alpha_{i1})ds] \right]$$
(18)

where $\Omega$ is a predefined compact set containing origin. By viewing $x_{i2}$ as optimal control $\alpha_{i1}^*$, the HJB equation linked with Eqs. (15) and (17) can be rewritten

$$H_{i1}\left(s_{i1}, \alpha_{i1}^*, \frac{dJ_{i1}^*(s_{i1})}{ds_{i1}}\right) = s_{i1}^2 + \alpha_{i1}^2 + \frac{dJ_{i1}^*}{ds_{i1}}\left(d_i\alpha_{i1}^* + F_{i1} - \sum_{j=1}^{N} a_{ij}x_{j2}\right) + \frac{1}{2}\frac{d^2 f_{i1}^*}{ds_{i1}^2}\Psi_{i1}^T\Psi_{i1} = 0 \quad (19)$$

The optimal virtual controller $\alpha_{i1}^*$ can be derived by solving $\partial H_{i1}/\partial \alpha_{i1}^* = 0$ as

$$\alpha_{i1}^* = -\frac{1}{2}\frac{dJ_{i1}^*(s_{i1})}{ds_{i1}} \tag{20}$$

To attain the tracking control, the term $\frac{dJ_{i1}^*(s_{i1})}{ds_{i1}}$ is partitioned as

$$\frac{dJ_{i1}^*(s_{i1})}{ds_{i1}} = \frac{2\gamma_{i1}}{d_i}s_{i1} + \frac{1}{2\beta_{i1}d_i}s_{i1}^3 + \frac{2}{d_i}h_{i1}(x_{i1},s_{i1}) + \frac{1}{d_i}J_{i1}^0(x_{i1},s_{i1}) \tag{21}$$

where $\gamma_{i1} > 0, \beta_{i1} > 0$ are two designed constants, $h_{i1}(x_{i1},s_{i1}) = F_{i1} + s_{i1}\|\Psi_{i1}\|^4$ and $J_{i1}^0(x_{i1},s_{i1}) = -\frac{2\gamma_{i1}}{d_i}s_{i1} - \frac{1}{2\beta_{i1}d_i}s_{i1}^3 - \frac{2}{d_i}h_{i1}(x_{i1},s_{i1}) + \frac{dJ_{i1}^*(s_{i1})}{ds_{i1}} \in \mathbb{R}$. Substituting Eqs. (21) into (20) yields

$$\alpha_{i1}^* = \frac{1}{d_i}[-\gamma_{i1}s_{i1} - \frac{1}{4\beta_{i1}}s_{i1}^3 - h_{i1}(x_{i1},s_{i1}) - \frac{1}{2}J_{i1}^0(x_{i1},s_{i1})] \tag{22}$$

Since two functions $h_{i1}(x_{i1},s_{i1})$ and $J_{i1}^0(x_{i1},s_{i1})$ are uncertain yet continuous, they can be approximated by NN as

$$h_{i1}(x_{i1},s_{i1}) = W_{hi1}^{*T}S_{hi1}(x_{i1},s_{i1}) + \varepsilon_{hi1}(x_{i1},s_{i1}) \tag{23}$$

$$J_{i1}^0(x_{i1},s_{i1}) = W_{Ji1}^T S_{Ji1}(x_{i1},s_{i1}) + \varepsilon_{Ji1}(x_{i1},s_{i1}) \tag{24}$$

where $W_{hi1}^{*T} \in \mathbb{R}^{p_1}$ and $W_{Ji1}^{*T} \in \mathbb{R}^{q_1}$ are the ideal NN weights, $S_{hi1}(x_{i1},s_{i1}) \in \mathbb{R}^{p_1}$ and $S_{Ji1}(x_{i1},s_{i1}) \in \mathbb{R}^{q_1}$ are basis function vectors, and $\varepsilon_{hi1}(x_{i1},s_{i1}) \in \mathbb{R}$, $\varepsilon_{Ji1}(x_{i1},s_{i1}) \in \mathbb{R}$ denote approximation errors. Substitute Eqs. (23) and (24) into Eqs. (21) and (22), separately

$$\frac{df_{i1}^*(s_{i1})}{ds_{i1}} = \frac{1}{d_i}[2\gamma_{i1}s_{i1}(t) + \frac{1}{2\beta_{i1}}s_{i1}^3(t) + 2W_{hi1}^{*T}S_{hi1}(x_{i1},s_{i1}) + W_{ji1}^{*T}S_{ji1}(x_{i1},s_{i1}) + \varepsilon_{i1}] \tag{25}$$

$$\alpha_{i1}^* = \frac{1}{d_i}[-\gamma_{i1}s_{i1}(t) - \frac{1}{4\beta_{i1}}s_{i1}^3(t) - W_{hi1}^{*T}S_{hi1}(x_{i1},s_{i1}) - \frac{1}{2}W_{Ji1}^{*T}S_{Ji1}(x_{i1},s_{i1}) - \frac{1}{2}\varepsilon_{i1}] \tag{26}$$

where $\varepsilon_{i1} = 2\varepsilon_{hi1} + \varepsilon_{Ji1}$. The optimal control Eq. (26) is unattainable due to the two ideal weights $W_{hi1}^{*T}$ and $W_{Ji1}^{*T}$ are uncertain constant vectors.

To acquire an effective optimized virtual control, the implementation involves applying RL through the identifier-critic-actor architecture, utilizing the NNs. The uncertain function $h_{i1}(x_{i1},s_{i1})$ of adaptive identifier is constructed in the following:

$$\hat{h}_{i1}(x_{i1},s_{i1}) = \hat{W}_{hi1}^T(t)S_{hi1}(x_{i1},s_{i1}) \tag{27}$$

where $\hat{h}_{i1}(x_{i1},s_{i1})$ is the identifier output, and $\hat{W}_{hi1}^T(t) \in \mathbb{R}^{p_1}$ is the NN weight. The weight experiences updates based on the following law:

$$\dot{\hat{W}}_{hi1}(t) = \Gamma_{i1}(S_{hi1}(x_{i1},s_{i1})s_{i1}^3(t) - \sigma_{i1}\hat{W}_{hi1}(t)) \tag{28}$$

where $\Gamma_{i1}$ is a positive-definite constant matrix, $\sigma_{i1} > 0$ is constant. Designing the critic to evaluate the control performance aligns with Eq. (25) as

$$\frac{d\hat{J}_{i1}^*(s_{i1})}{ds_{i1}} = \frac{1}{d_i}\left[2\gamma_{i1}s_{i1}(t) + \frac{1}{2\beta_{i1}}s_{i1}^3(t) + 2\hat{W}_{hi1}^T(t)S_{hi1}(x_{i1},s_{i1}) + \hat{W}_{ci1}^T(t)S_{Ji1}(x_{i1},s_{i1})\right] \tag{29}$$

where $\frac{d\hat{J}_{i1}^*(s_{i1})}{ds_{i1}} \in \mathbb{R}$ is the estimation of $\frac{dJ_{i1}^*(s_{i1})}{ds_{i1}}$, $\hat{W}_{ci1}^T \in \mathbb{R}^{q_1}$ is the NN weight of critic. The weight experiences updates based on the following law:

$$\dot{\hat{W}}_{ci1}(t) = -\gamma_{ci1}S_{Ji1}(x_{i1}, s_{i1})S_{Ji1}^T(x_{i1}, s_{i1})\hat{W}_{ci1}(t) \tag{30}$$

where $\gamma_{ci1} > 0$ is constant. The formulation of the actor, responsible for executing the control action, corresponds to Eq. (25) as articulated below:

$$\hat{\alpha}_{i1}^* = \frac{1}{d_i}[-\gamma_{i1}s_{i1}(t) - \frac{1}{4\beta_{i1}}s_{i1}^3(t) - \hat{W}_{hi1}^T(t)S_{hi1}(x_{i1}, s_{i1}) - \frac{1}{2}\hat{W}_{ai1}^T(t)S_{Ji1}(x_{i1}, s_{i1})] \tag{31}$$

where $\hat{\alpha}_{i1}^*$ is the optimized virtual control, $\hat{W}_{ai1}^T(t) \in \mathbb{R}^{q_1}$ is the NN weight of actor. The weight experiences updates based on the following law:

$$\hat{W}_{ai1}(t) = -S_{Ji1}(x_{i1}, s_{i1})S_{Ji1}^T(x_{i1}, s_{i1}) \times \left(\gamma_{ai1}(\hat{W}_{ai1}(t) - \hat{W}_{ci1}(t)) + \gamma_{ci1}\hat{W}_{ci1}(t)\right) \tag{32}$$

where $\gamma_{ai1} > 0$ is constant. These determined parameters, $\beta_{i1}$, $\gamma_{i1}$, $\gamma_{ci1}$, and $\gamma_{ai1}$, are selected to satisfy

$$\beta_{i1} > 0, \gamma_{i1} > 3, \gamma_{ai1} > \frac{\beta_{i1}}{2}, \gamma_{ai1} > \gamma_{ci1} > \frac{\gamma_{ai1}}{2} \tag{33}$$

According to Eqs. (19), (29) and (31), the HJB equation is calculated as

$$\begin{aligned}
H_{i1}(s_{i1}, \hat{\alpha}_{i1}^*, \frac{d\hat{J}_{i1}^*}{ds_{i1}}) &= s_{i1}^2(t) + \frac{1}{d_i^2}(-\gamma_{i1}s_{i1}(t) - \frac{1}{4\beta_{i1}}s_{i1}^3(t) - \hat{W}_{hi1}^T(t)S_{hi1}(x_{i1}, s_{i1}) \\
&- \frac{1}{2}\hat{W}_{ai1}^T(t)S_{Ji1}(x_{i1}, s_{i1}))^2 + \frac{1}{d_i^2}[2\gamma_{i1}s_{i1}(t) + \frac{1}{2\beta_{i1}}s_{i1}^3(t) + 2\hat{W}_{hi1}^T(t)S_{hi1}(x_{i1}, s_{i1}) \\
&+ \hat{W}_{ci1}^T(t)S_{Ji1}(x_{i1}, s_{i1})] \times (-\gamma_{i1}s_{i1}(t) - \frac{1}{4\beta_{i1}}s_{i1}^3(t) - \hat{W}_{hi1}^T(t)S_{hi1}(x_{i1}, s_{i1}) - \frac{1}{2}\hat{W}_{ai1}^T(t) \times \\
&S_{Jil}(x_{i1}, s_{i1}) + f_{i1}(x_{i1}) + \psi_{i1}^T(x_{i1})\frac{dw}{dt} - \dot{y}_d) + \frac{1}{2}\frac{d^2J_{i1}^*}{ds_{i1}^2} \parallel \psi_{i1}(x_{i1}) \parallel^2
\end{aligned} \tag{34}$$

Building upon the preceding analysis, the optimized control $\hat{\alpha}_{i1}^*$ is foreseen as the sole solution to achieve $H_{i1}(s_{i1}, \hat{a}_{i1}^*, (d\hat{J}_{i1}^*)/(ds_{i1})) \to 0$. Assuming the existence of $H_{i1}\left(s_{i1}, \hat{a}_{i1}^*, \frac{d\hat{J}_{i1}^*}{ds_{i1}}\right) = 0$ and its unique solution, it is equivalent to the following equation:

$$\frac{\partial H_{i1}(s_{i1}, \hat{a}_{i1}^*, \frac{d\hat{J}_{i1}^*}{ds_{i1}})}{\partial \hat{W}_{ai1}} = \frac{1}{2}S_{Ji1}(x_{i1}, s_{i1})S_{Ji1}^T(x_{i1}, s_{i1}) \times \left(\hat{W}_{ai1}(t) - \hat{W}_{ci1}(t)\right) = 0 \tag{35}$$

Define the positive function $P_{i1}(t)$ as

$$P_{i1}(t) = \left(\hat{W}_{ai1}(t) - \hat{W}_{ci1}(t)\right)^T(\hat{W}_{ai1}(t) - \hat{W}_{ci1}(t)) \tag{36}$$

It is evident that Eq. (35) is the equivalent to $P_{i1}(t) = 0$. Given the fact that $(\partial P_{i1}(t))/(\partial \hat{W}_{ai1}(t)) = -(\partial P_{i1}(t))/(\partial \hat{W}_{ci1}(t)) = 2(\hat{W}_{ai1}(t) - \hat{W}_{ci1}(t))$, the time derivative of $P_{i1}(t)$ along with Eqs. (29) and (31) is

$$\begin{aligned}
\frac{dP_{i1}}{dt} &= \frac{\partial P_{i1}}{\partial \hat{W}_{ai1}^T}\dot{\hat{W}}_{ai1} + \frac{\partial P_{i1}}{\partial \hat{W}_{ci1}^T}\dot{\hat{W}}_{ci1} = -\frac{\partial P_{i1}}{\partial \hat{W}_{ai1}^T}S_{Ji1}S_{Ji1}^T(\gamma_{ai1}(\hat{W}_{ai1} - \hat{W}_{ci1}) + \gamma_{ci1}\hat{W}_{ci1}) \\
&= -\gamma_{ai1}\frac{\partial P_{i1}}{\partial \hat{W}_{ai1}^T}S_{Ji1}S_{Ji1}^T(\hat{W}_{ai1} - \hat{W}_{ci1}) = -\frac{\gamma_{ai1}}{2}\frac{\partial P_{i1}}{\partial \hat{W}_{ai1}^T}S_{Ji1}S_{Ji1}^T\frac{\partial P_{i1}}{\partial \hat{W}_{ai1}} \leq 0
\end{aligned} \tag{37}$$

The inequality Eq. (37) suggests that the updating laws Eqs. (30) and (32) can ensure eventually. The key benefits of the RL design approach include: (1) comparatively, the optimized control algorithm demonstrates a substantially simpler structure than existing optimal methods, such as *Vamvoudakis & Lewis (2010)*, *Liu et al. (2013)*, *Wen, Ge & Tu (2018)*. (2) this can alleviate the necessity for persistent excitation, a requirement prevalent in many optimal control methods. Replace $x_{i2}$ with $\alpha_{i1}^* + s_{i2}$ in the dynamic Eq. (14) to have

$$ds_{i1} = \left[ d_i(\alpha_{i1}^* + s_{i2}) + F_{i1} - \sum_{j=1}^{N} a_{ij}x_{j2} \right] dt + \Psi_{i1} dw \tag{38}$$

The Lyapunov function candidate is designed as

$$L_{i1} = \frac{1}{4}s_{i1}^4 + \frac{1}{2}W_{hi1}^T \Gamma_{i1}^{-1} W_{hi1} + \frac{1}{2}W_{ci1}^T W_{ci1} + \frac{1}{2}W_{ai1} W_{ai1} \tag{39}$$

where $\tilde{W}_{hi1}(t) = \hat{W}_{hi1}(t) - W_{hi1}^*$, $\tilde{W}_{ci1}(t) = \hat{W}_{ci1}(t) - W_{Ji1}^*$ and $\tilde{W}_{ai1}(t) = \hat{W}_{ai1}(t) - W_{Ji1}^*$ represent corresponding errors. Compute the $\mathfrak{L}$ of $L_{i1}$, along with Eqs. (28), (30), (32) and (39) to yield

$$\mathcal{L}L_{i1} = s_{i1}^3 \left[ d_i(\alpha_{i1}^* + s_{i2}) + F_{i1} - \sum_{j=1}^{N} a_{ij}x_{j2} \right] + \frac{3}{2}s_{i1}^2 \parallel \Psi_{i1} \parallel^2 + \tilde{W}_{hi1}^T(S_{hi1}s_{i1}^3 - \sigma_{i1}\hat{W}_{hi1}) -$$
$$\gamma_{ci}\hat{W}_{ci1}^T S_{Ji1} S_{Ji1}^T \hat{W}_{ci1} - \tilde{W}_{ai1}^T S_{Ji1} S_{Ji1}^T [\gamma_{ai1}(\hat{W}_{ai1} - \hat{W}_{ci1}) + \gamma_{ci1}\hat{W}_{ci1}] \tag{40}$$

Design optimal virtual controller

$$\hat{\alpha}_{i1}^* = \frac{1}{d_i}(-\gamma_{i1}s_{i1} - \frac{1}{4\beta_{i1}}s_{i1}^3 - \hat{W}_{hi1}^T S_{hi1} - \frac{1}{2}\hat{W}_{ai1}^T S_{Ji1}) \tag{41}$$

and then Eq. (31) becomes

$$\mathcal{L}L_{i1} = s_{i1}^3 \left[ \gamma_{i1}s_{i1} - \frac{1}{4\beta_{i1}}s_{i1}^3 - \hat{W}_{hi1}^T S_{hi1} - \frac{1}{2}\hat{W}_{ai1} S_{Ji1} + s_{i2}d_i + F_{i1} - \sum_{j=1}^{N} a_{ij}x_{j2} \right] + \frac{3}{2}s_{i1}^2 \parallel \Psi_{i1} \parallel^2$$
$$+ \tilde{W}_{hi1}^T(s_{hi1}s_{i1}^3 - \sigma_{i1}\hat{W}_{hi1}) - \gamma_{ci1}\tilde{W}_{ci1}^T S_{Ji1} S_{Ji1}^T \hat{W}_{ci1} - \gamma_{ai1}\tilde{W}_{ai1}^T S_{Ji1} S_{Ji1}^T \hat{W}_{ai1} + (\gamma_{ai1} - \gamma_{ci1})$$
$$\tilde{W}_{ai1}^T S_{Ji1} S_{Ji1}^T \hat{W}_{ci1} \tag{42}$$

With Young's inequality Eq. (8), there are following results:

$$d_i s_{i1}^3 s_{i2} \le \frac{3}{4}d_i s_{i1}^4 + \frac{1}{4}d_i s_{i2}^4 \tag{43}$$

$$-s_{i1}^3 \sum_{j=1}^{N} a_{ij}x_{j2} \le \frac{3}{4}s_{i1}^4 + \frac{1}{4}\left( \sum_{j=1}^{N} a_{ij}x_{j2} \right)^4 \tag{44}$$

$$\frac{3}{2}s_{i1}^2 ||\Psi_{i1}||^2 \le s_{i1}^4 ||\Psi_{i1}||^4 + \frac{9}{16} \tag{45}$$

$$-\frac{1}{2}s_{i1}^3 \hat{W}_{ai1}^T S_{Ji1} \le \frac{1}{4\beta_{i1}}s_{i1}^6 + \frac{\beta_{i1}}{4}\hat{W}_{ai1} S_{Ji1} S_{Ji1}^T \hat{W}_{ai1} \quad (46)$$

Substituting inequalities Eqs. (43), (44), (45) and (46) into (42) has

$$\mathcal{L}L_{i1} \le -(\gamma_{i1} - \frac{3}{4}d_i - \frac{3}{4})s_{i1}^4 - s_{i1}^3(\hat{W}_{hi1}^T S_{hi1} - h_{i1}) + W_{hi1}^T(S_{hi1}s_{i1}^3 - \sigma_{i1}\hat{W}_{hi1}) -$$
$$\gamma_{ci1}\widetilde{W}_{ci1}^T S_{Ji1} S_{Ji1}^T \hat{W}_{ci1} - \gamma_{ai1}\widetilde{W}_{ai1}^T S_{Ji1} S_{Ji1}^T \hat{W}_{ai1} + (\gamma_{ai1} - \gamma_{ci1})\widetilde{W}_{ai1}^T S_{Ji1} S_{Ji1}^T \hat{W}_{ci1} +$$
$$\frac{\beta_{i1}}{4}\hat{W}_{ai1}^T S_{Ji1} S_{Ji1}^T \hat{W}_{ai1} + \frac{1}{4}(\sum_{j=1}^N a_{ij}x_{j2})^4 + \frac{9}{16} + \frac{1}{4}d_i s_{i2}^4 \quad (47)$$

where $h_{i1} = F_{i1} + s_{i1}||\Psi_{i1}||^4$. Substituting Eqs. (24) into (47) results in the following inequality:

$$\mathcal{L}L_{i1} \le -\left(\gamma_{i1} - \frac{3}{4}d_i - \frac{3}{4}\right)s_{i1}^4 + s_{i1}^3\varepsilon_{hi} - \sigma_{i1}\widetilde{W}_{hi1}^T \hat{W}_{hi1} - \gamma_{ci1}\widetilde{W}_{ci1}^T S_{Ji1} S_{Ji1}^T \hat{W}_{ci1} -$$
$$\gamma_{ai1}\widetilde{W}_{ai1}^T S_{Ji1} S_{Ji1}^T \hat{W}_{ai1} + \frac{\beta_{i1}}{4}\hat{W}_{ai1}^T S_{Ji1} S_{Ji1}^T \hat{W}_{ai1} + (\gamma_{ai1} - \gamma_{ci1})\widetilde{W}_{ai1}^T S_{Ji1} S_{Ji1}^T \hat{W}_{ci1} +$$
$$\frac{1}{4}(\sum_{i=1}^N a_{ji}x_{j2})^4 + \frac{1}{4}d_i s_{i2}^4 + \frac{9}{16} \quad (48)$$

From the facts $\tilde{W}_{hi1}(t) = \hat{W}_{hi1}(t) - W_{hi1}^*$, $\tilde{W}_{ci1}(t) = \hat{W}_{ci1}(t) - W_{Ji1}^*$ and $\tilde{W}_{ai1}(t) = \hat{W}_{ai1}(t) - W_{Ji1}^*$, the following equations can be derived:

$$\widetilde{W}_{hi1}^T \hat{W}_{hi1} = \frac{1}{2}\widetilde{W}_{h1i}^T \widetilde{W}_{hi1} + \frac{1}{2}\hat{W}_{hi1}^T \hat{W}_{hi1} - \frac{1}{2}W_{hi1}^{*T} W_{hi1}^* \quad (49)$$

$$\widetilde{W}_{ci1}^T S_{Ji1} S_{Ji1}^T \hat{W}_{ci1} = \frac{1}{2}\widetilde{W}_{ci1}^T S_{Ji1}^T \widetilde{W}_{ci1} + \frac{1}{2}\hat{W}_{ci1}^T S_{Ji1}^T \hat{W}_{ci1} - \frac{1}{2}W_{Ji1}^{*T} S_{Ji1} S_{Ji1}^T W_{Ji1}^* \quad (50)$$

$$\widetilde{W}_{ai1}^T S_{Ji1} S_{Ji1}^T \hat{W}_{ai1} = \frac{1}{2}\widetilde{W}_{ai1}^T S_{Ji1} S_{Ji1}^T \widetilde{W}_{ai1} + \frac{1}{2}\hat{W}_{ai1}^T S_{Ji1} S_{Ji1}^T \hat{W}_{ai1} - \frac{1}{2}W_{Ji1}^{*T} S_{Ji1} S_{Ji1}^T W_{Ji1}^* \quad (51)$$

With Young's inequality Eqs. (8) and limitation of (33), subsequent inequalities obtained:

$$s_{i1}^3\varepsilon_{hi1} \le \frac{3}{4}s_{i1}^4 + \frac{1}{4}\varepsilon_{hi1}^4 \quad (52)$$

$$(\gamma_{ai1} - \gamma_{ci1})\widetilde{W}_{ai1}^T S_{Ji1} S_{Ji1}^T \hat{W}_{ci1} \le \frac{\gamma_{ai1} - \gamma_{ci1}}{2}\widetilde{W}_{ai1}^T S_{Ji1} S_{Ji1}^T \widetilde{W}_{ai1} + \frac{\gamma_{ai1} - \gamma_{ci1}}{2}\hat{W}_{ci1}^T S_{Ji1} S_{Ji1}^T \hat{W}_{ci1} \quad (53)$$

Substituting Eqs. (49)–(53) into (48) yields

$$\mathcal{L}L_{i1} \le -\left(\gamma_{i1} - \frac{3}{2} - \frac{3}{4}d_i\right)s_{i1}^4 - \frac{\sigma_{i1}}{2}\widetilde{W}_{hi1}^T \widetilde{W}_{hi1} - \frac{\gamma_{ci1}}{2}\widetilde{W}_{ci1}^T S_{Ji1} S_{Ji1}^T \widetilde{W}_{ci1} - \frac{\gamma_{ci1}}{2}\widetilde{W}_{ai1}^T S_{Ji1} S_{Ji1}^T \widetilde{W}_{ai1}$$
$$-(\gamma_{ci1} - \frac{\gamma_{ai1}}{2})(\hat{W}_{ci1}^{*T} S_{Ji1})^2 - (\frac{\gamma_{ai1}}{2} - \frac{\beta_{i1}}{4})(\hat{W}_{ai1}^T S_{Ji1})^2 + B_{i1} + \frac{d_i}{4}s_{i2}^4 + \frac{1}{4}(\sum_{j=1}^N a_{ij}x_{j2})^4 \quad (54)$$

where $B_{i1}(t) = (\frac{\gamma_{ci1}}{2} + \frac{\gamma_{ai1}}{2})(W_{Ji1}^{*T} S_{Ji1})^2 + \frac{\sigma_{i1}}{2}||W_{hi1}^{*}||^2 + \frac{1}{4}\varepsilon_{hi1}^4 + \frac{9}{16}$ and $|B_{i1}(t)| \leq b_{i1}$, because all its terms are bounded, and $\frac{1}{4}\left(\sum_{j=1}^{N} a_{ij}x_{j2}\right)^4$ will be handled in step 2's $h_{i2}(x_{i2}, s_{i2})$.

Step m $(2 \leq m \leq n-1)$: Define the containment error as $s_{im} = x_{im} - \hat{\alpha}_{im-1}^{*}$. According to Eq. (9), the error dynamic, along with Eq. (13), is

$$ds_{im} = [x_{im+1} + f_{im}(x_{im}) - \mathcal{L}\hat{\alpha}_{im-1}^{*}]dt + \Psi_{im}dw \tag{55}$$

where $\Psi_{im} = \psi_{im}(\overline{x}_{im}) - \sum_{j=1}^{m-1} \frac{\partial \hat{\alpha}_{im-1}^{*}}{\partial x_{ij}}\psi_{ij}$. Let $\alpha_{im}$ denote virtual controller, the performance index function can be defined as

$$J_{im}(s_{im}) = \int_{t}^{\infty} c_{im}(s_{im}(s), \alpha_{im}(s_{im}(s)))ds \tag{56}$$

where $c_{im}(s_{im}, \alpha_{im}) = s_{im}^2(t) + \alpha_{im}^2$ is the cost function. Denoted $\alpha_{im}^{*}$ as the optimal virtual controller, substitute $\alpha_{im}^{*}$ into Eq. (56), the function can be rewritten as

$$J_{im}^{*}(s_{im}) = \int_{t}^{\infty} c_{im}(s_{im}(s), \alpha_{im}^{*}(s_{im}(s)))ds. \tag{57}$$

Similar to Step 1, Eq. (57) manifests the subsequent characteristic

$$\mathbb{E}[J_{im}^{*}(s_{im})] = \min_{\alpha_{im} \in \Psi(\Omega)} \mathbb{E}[J_{im}(s_{im})]. \tag{58}$$

By viewing $x_{im+1}(t)$ as optimal control $\alpha_{im}^{*}$, the HJB equation relate to Eqs. (55) and (57) is

$$H_{im}\left(s_{im}, \alpha_{im}^{*}, \frac{dJ_{im}^{*}}{ds_{im}}\right) = s_{im} + \alpha_{im}^{*}2 + \frac{dJ_{im}^{*}}{ds_{im}} \times \left(\alpha_{im}^{*} + f_{im} + \Psi_{im}\frac{dw}{dt} - \mathcal{L}\hat{\alpha}_{im-1}^{*}\right) +$$
$$\frac{1}{2}\frac{d^2 J_{im}^{*}}{ds_{im}^2}\Psi_{im}^{T}\Psi_{im} = 0 \tag{59}$$

where $(dw)/(dt)$ represents the white noise. Besides, $\alpha_{im}^{*}$ is obtained by solving $(\partial H_{im})/(\partial \alpha_{im}^{*}) = 0$ as

$$\alpha_{im}^{*} = -\frac{1}{2}\frac{dJ_{im}^{*}}{ds_{im}} \tag{60}$$

To attain the containment control, the term $(dJ_{im}^{*}(s_{im}))/(ds_{im})$ is segmented as

$$\frac{dJ_{im}^{*}}{ds_{im}} = 2\gamma_{im}s_{im} + \frac{1}{2\beta_{im}}s_{im}^3 + 2h_{im} + J_{im}^0 \tag{61}$$

where $\gamma_{im} > 0$ and $\beta_{im} > 0$ are two designed constants, $h_{i2} = f_{i2} + s_{i2}||\Psi_{im}||^4 - \frac{1}{4}\left(\sum_{j=1}^{N} a_{ij}x_{j2}\right)^4 \in \mathbb{R}$, $h_{im} = f_{im} + s_{im}||\Psi_{im}||^4 \in \mathbb{R}(m \geq 3)$, and $J_{im}^0 = -2\gamma_{im}s_{im} - \frac{1}{2\beta_{im}}s_{im}^3 - 2h_{im} + \frac{dJ_{im}^{*}}{ds_{im}} \in \mathbb{R}$. By substituting Eqs. (61) into (60), optimal control transforms into

$$\alpha_{im}^{*} = -\gamma_{im}s_{im} - \frac{1}{4\beta_{im}}s_{im}^3 - h_{im} - J_{im}^0 \tag{62}$$

Since two functions $h_{im}(\overline{x}_{im}, s_{im})$ and $J_{im}^0(\overline{x}_{im}, s_{im})$ are uncertain yet continuous, they can be approximated by NN as

$$h_{im}(\overline{x}_{im}, s_{im}) = W_{him}^{*T} S_{him}(\overline{x}_{im}, s_{im}) + \varepsilon_{him}(\overline{x}_{im}, s_{im}) \tag{63}$$

$$J_{im}^0(\overline{x}_{im}, s_{im}) = W_{Jim}^T S_{Jim}(\overline{x}_{im}, z_{im}) + \varepsilon_{Jim}(\overline{x}_{im}, s_{im}) \tag{64}$$

where $W_{him}^{*T} \in \mathbb{R}^{p_m}$ and $W_{Jim}^{*T} \in \mathbb{R}^{q_m}$ are the ideal NN weights, $S_{him}(x_{im}, s_{im}) \in \mathbb{R}^{p_m}$, $S_{Jim}(x_{im}, s_{im}) \in \mathbb{R}^{q_m}$ are basis vectors, $\varepsilon_{him}(x_{im}, s_{im}) \in \mathbb{R}, \varepsilon_{Jim}(x_{im}, s_{im}) \in \mathbb{R}$ are bounded approximation errors. Substituting Eqs. (63) and (64) into Eqs. (61) and (62) has

$$\frac{dJ_{im}^*(s_{im})}{ds_{im}} = 2\gamma_{im}s_{im} + \frac{1}{2\beta_{im}}s_{im}^3 + 2W_{him}^{*T}S_{him}(\overline{x}_{im}, s_{im}) + W_{Jim}^{*T}S_{Jim}(\overline{x}_{im}, s_{im}) + \varepsilon_{im} \tag{65}$$

$$\alpha_{im}^* = -\gamma_{im}s_{im} - \frac{1}{4\beta_{im}}s_{im}^3 - W_{him}^{*T}S_{him} - \frac{1}{2}W_{Jim}^{*T}S_{Jim} - \frac{1}{2}\varepsilon_{im} \tag{66}$$

where $\varepsilon_{im} = 2\varepsilon_{him} + \varepsilon_{Jim}$. The optimal control Eq. (66) is impractical due to the two ideal weights $W_{him}^{*T}$ and $W_{Jim}^{*T}$ are uncertain. To obtain a practical optimized control, RL is constructed based on Eqs. (65) and (66) as follows. The adaptive identifier is formulated as follows:

$$\hat{h}_{im}(\overline{x}_{im}, s_{im}) = \hat{W}_{him}^T S_{him}(\overline{x}_{im}, s_{im}) \tag{67}$$

where $\hat{h}_{im}(x_{im}, s_{im})$ is the identifier output, $\hat{W}_{him}^T(t) \in \mathbb{R}^{p_m}$ is the NN weight. The weight experiences updates based on the following law:

$$\dot{\hat{W}}_{him} = \Gamma_{im}(S_{him}(\overline{x}_{im}, s_{im})s_{im}^3 - \sigma_{im}\hat{W}_{him}) \tag{68}$$

where $\Gamma_{im}$ is a positive-definite constant matrix, $\sigma_{im} > 0$ is constant. The critic is designed in the following:

$$\frac{d\hat{J}_{im}^*(s_{im})}{ds_{im}} = 2\gamma_{im}s_{im} + \frac{1}{2\beta_{im}}s_{im}^3 + 2\hat{W}_{him}^T S_{him} + \hat{W}_{cim}^T S_{Jim} \tag{69}$$

where $d\hat{J}_{im}^*(s_{im})/ds_{im} \in \mathbb{R}$ is the estimation of $dJ_{im}^*(s_{im})/ds_{im}$, $\hat{W}_{cim}^T(t) \in \mathbb{R}^{q_m}$ is the NN weight of critic. The weight experiences updates based on the following law:

$$\dot{\hat{W}}_{cim} = -\gamma_{cim}S_{Jim}S_{Jim}^T\hat{W}_{cim} \tag{70}$$

where $\gamma_{cim} > 0$ is constant. The actor is designed as follows:

$$\hat{\alpha}_{im}^* = -\gamma_{im}s_{im} - \frac{1}{4\beta_{im}}s_{im}^3 - \hat{W}_{him}^T S_{him} - \frac{1}{2}\hat{W}_{aim}S_{Jim} \tag{71}$$

where $\hat{\alpha}_{im}^*$ is the optimized virtual control, $\hat{W}_{aim}^T(t) \in \mathbb{R}^{q_m}$ is the NN weight of actor. The weight experiences updates based on the following law:

$$\dot{\hat{W}}_{aim} = -S_{Jim}S_{Jim}^T(\gamma_{aim}(\hat{W}_{aim} - \hat{W}_{cim}) + \gamma_{cim}\hat{W}_{cim}) \tag{72}$$

where $\gamma_{aim} > 0$ are constant. These designed parameters, $\beta_{im}, \gamma_{im}, \gamma_{cim}$ and $\gamma_{aim}$ satisfy the following conditions:

$$\beta_{im} > 0, \gamma_{im} > 4, \gamma_{aim} > \frac{\beta_{im}}{2}, \gamma_{aim} > \gamma_{cim} > \frac{\gamma_{aim}}{2}. \tag{73}$$

Define containment error of the step m+1 as $s_{im+1} = x_{im+1} - \alpha^*_{im+1}$. Replace $x_{im+1}$ with $\alpha^*_{im+1} + s_{im+1}$ in the dynamic Eq. (55) to have

$$ds_{im} = (\hat{\alpha}^*_{im} + s_{im+1} + f_{im} - \mathcal{L}\hat{\alpha}^*_{im-1})dt + \Psi_{im}dw \tag{74}$$

Select the Lyapunov function candidate:

$$L_{im} = \sum_{j=1}^{m-1} L_{ij} + \frac{1}{4}s_{im}^4 + \frac{1}{2}\widetilde{W}_{him}^T\widetilde{W}_{him} + \frac{1}{2}\widetilde{W}_{cim}^T\widetilde{W}_{cim} + \frac{1}{2}\widetilde{W}_{aim}^T\widetilde{W}_{aim} \tag{75}$$

where $L_{ij} = \frac{1}{4}s_{ij}^4 + \frac{1}{2}\widetilde{W}_{hij}^T\Gamma_{ij}^{-1}\widetilde{W}_{hij} + \frac{1}{2}\widetilde{W}_{cij}^T\widetilde{W}_{cij} + \frac{1}{2}\widetilde{W}_{aij}^T\widetilde{W}_{aij}$, and $\widetilde{W}_{him}(t) = \hat{W}_{him}(t) - W^*_{him}$, $\widetilde{W}_{cim}(t) = \hat{W}_{cim}(t) - W^*_{Jim}$ and $\widetilde{W}_{aim}(t) = \hat{W}_{aim}(t) - W^*_{Jim}$. Computing the infinitesimal generator $\mathfrak{L}$ of $L_{im}$, along with Eqs. (68), (70), (72) and (74) has

$$\mathcal{L}L_{im} = \sum_{j=1}^{m-1}\mathcal{L}L_{ij} + s_{im}^3(\hat{\alpha}^*_{im} + s_{im+1} + f_{im} - \mathcal{L}\hat{\alpha}^*_{im-1}) + \frac{3}{2}s_{im}^2||\Psi_{im}||^2 + \widetilde{W}_{him}^T(S_{him}s_{im}^3 - \sigma_{im}\hat{W}_{him})$$
$$-\gamma_{cim}\widetilde{W}_{cim}^T S_{Jim}S_{Jim}^T\hat{W}_{cim} - \widetilde{W}_{ain}^T S_{Jim}S_{Jim}^T[\gamma_{aim}(\hat{W}_{aim} - \hat{W}_{cim}) + \gamma_{cim}\hat{W}_{cim}] \tag{76}$$

Substituting the virtual control Eqs. (71) into (76) holds

$$\mathcal{L}L_{im} = \sum_{i=1}^{m-1}\mathcal{L}L_{ij} + s_{im}^3\left(-\gamma_{im}s_{im} - \frac{1}{4\beta_{im}}s_{im}^3 - \hat{W}_{him}^T S_{him} - \frac{1}{2}\hat{W}_{aim}^T S_{Jim} + s_{im+1} + f_{im} - \right.$$
$$\mathcal{L}\hat{\alpha}^*_{im-1}\right) + \frac{3}{2}s_{im}^2||\Psi_{im}||^2 + \widetilde{W}_{him}^T(S_{him}s_{im}^3 - \sigma_{im}\hat{W}_{him}) - \gamma_{cim}\widetilde{W}_{cim}^T S_{Jim}S_{Jim}^T\hat{W}_{cim}$$
$$-\widetilde{W}_{ain}^T S_{Jim}S_{Jim}^T[\gamma_{aim}(\hat{W}_{aim} - \hat{W}_{cim}) + \gamma_{cim}\hat{W}_{cim}] \le \sum_{j=1}^{m-1}\mathcal{L}L_{ij} - \left(\gamma_{im} - \frac{3}{2}\right)s_{im}^4 +$$
$$\frac{\beta_{im}}{4}\hat{W}_{aim}^T S_{Jim}S_{Jim}^T\hat{W}_{aim} + \frac{1}{4}s_{im+1}^4 + \frac{1}{4}\mathcal{L}\hat{\alpha}^*_{im+1} + \frac{9}{16} - \widetilde{W}_{him}^T(S_{him}s_{im}^3 - h_{im})$$
$$+\widetilde{W}_{him}^T(S_{him}s_{im}^3 - \sigma_{im}\hat{W}_{him}) - \gamma_{cim}\widetilde{W}_{cim}^T S_{Jim}S_{Jim}^T\hat{W}_{cim} - \gamma_{aim}\widetilde{W}_{aim}^T S_{Jim}S_{Jim}^T\hat{W}_{aim} +$$
$$(\gamma_{aim} - \gamma_{cim})\widetilde{W}_{aim}^T S_{Jim}S_{Jim}^T\widetilde{W}_{cim} \tag{77}$$

From the fact $-s_{im}^3(t)\mathfrak{L}\hat{\alpha}^*_{im-1} \le (3/4)s_{im}^4(t) + (1/4)(\mathfrak{L}\hat{\alpha}^*_{im-1})^4$ and previous results, following numerous operations resembling those in Eqs. (43)–(54) in Step 1, (87) can be expressed as

$$\mathcal{L}L_{im} \le \sum_{j=1}^{m-1}(-a_{ij}L_{ij} + b_{ij}) - (\gamma_{im} - 4)s_{im}^2 - \frac{\sigma_{im}}{2\lambda_{\Gamma_{im}^{-1}}^{max}}\widetilde{W}_{him}^T\Gamma_{im}^{-1}\widetilde{W}_{him}$$
$$-\frac{\gamma_{cim}}{2}\lambda_{S_{Jim}}^{min}\widetilde{W}_{cim}^T\widetilde{W}_{cim} - \frac{\gamma_{cim}}{2}\lambda_{S_{Jim}}^{min}\widetilde{W}_{cim}^T\widetilde{W}_{cim} + B_{im} + \frac{1}{4}s_{im+1}^4 \tag{78}$$

where $\lambda_{\Gamma_{im}^{-1}}^{max}$ is the maximal eigenvalue of $\Gamma_{im}^{-1}$, $\lambda_{S_{Jim}}^{min}$ is the minimal eigenvalue of $S_{Jim}S_{Jim}^T$.

And $B_{im} = (\frac{\gamma_{cim}}{2} + \frac{\gamma_{aim}}{2})(W_{Jim}^{*T}S_{Jim})^2 + \frac{\sigma_{im}}{2}||W^*_{him}||^2 + \frac{1}{4}(\mathcal{L}\hat{\alpha}^*_{im-1})^4 + \frac{1}{4}\varepsilon_{him}^4 + \frac{9}{16}$ ,which

satisfied $|B_{im}| \le b_{im}$. Define $a_{im} = \min\left\{4(\gamma_{im} - 4), \frac{\sigma_{im}}{\lambda_{\Gamma_{im}^{-1}}^{max}}, \gamma_{cim}\lambda_{S_{Jim}}^{min}\right\}$, and then Eq. (78) can

become the following one:

$$\mathcal{L}L_{im} \leq \sum_{j=1}^{m}(-a_{ij}L_{ij}+b_{ij})+\frac{1}{4}s_{im+1}^4 \tag{79}$$

Step n: The optimized control $u_i$ is obtained here. Based on Eq. (9), $s_{in}=x_{in}-\hat{\alpha}_{in-1}^*$ can be derived from Eq. (13) as follows:

$$ds_{in}=(u_i+f_{in}(\overline{x}_{in})-\mathcal{L}\hat{\alpha}_{n-1}^*)dt+\Psi_{in}dw \tag{80}$$

where $\Psi_{in}=\psi_{in}-\sum_{j=1}^{n-1}\frac{\partial\alpha_{in-1}}{\partial x_{ij}}\psi_{ij}$. The performance index function related to Eq. (80) can be written as

$$J_{in}(s_{in})=\int_t^\infty c_{in}(s_{in}(s),u_i(s_{in}(s)))ds \tag{81}$$

where $c_{in}(s_{in},u_i)=\sin 2+u_i^2$ is cost function. Denoted $u_i^*$ as optimal control, the function can be rewritten as

$$J_{in}^*(s_{in})=\int_t^\infty c_{in}(s_{in}(s),u_i^*(s_{in}(s)))ds \tag{82}$$

The function Eq. (82) implies the following property:

$$\mathbb{E}[J_{in}^*(s_{in})]=\min_{u_i\in\Psi(\Omega)}\mathbb{E}[J_{in}(s_{in})] \tag{83}$$

The HJB equation related to Eqs. (80) and (82) is

$$H_{in}\left(s_{in},u_i^*,\frac{dJ_{in}^*}{ds_{in}}\right)=\sin 2+u_i^{*2}+\frac{dJ_{in}^*}{ds_{in}}\left(u_i^*+f_{jn}-\mathcal{L}\hat{\alpha}_{in-1}^*+\Psi_{in}\frac{dw}{dt}\right)+$$
$$\frac{1}{2}\frac{d^2J_{in}^*}{d\sin 2}\Psi_{in}^T\Psi_{in}=0 \tag{84}$$

Solving $(\partial H_{in})/(\partial u_i^*)=0$ yields

$$u_i^*=-\frac{1}{2}\frac{dJ_{in}^*(s_{in})}{ds_{in}} \tag{85}$$

Split the term $\frac{dJ_{in}^*}{ds_{in}}$ as

$$\frac{dJ_{in}^*}{ds_{in}}=2\gamma_{in}s_{in}+\frac{1}{2\beta_{in}}\sin 3+2h_{in}+J_{in}^0 \tag{86}$$

where $\gamma_{in}>0$ and $\beta_{in}>0$ are two designed constants, and $h_{in}=f_{in}+s_{in}||\Psi_{in}||^4\in\mathbb{R}$, $J_{in}^0=-2\gamma_{in}s_{in}-\frac{1}{2\beta_{in}}\sin 3-2h_{in}+\frac{dJ_{in}^*}{ds_{in}}\in\mathbb{R}$. Substituting Eqs. (86) into (85) has

$$u_i^*=-\gamma_{in}s_{in}-\frac{1}{4\beta_{in}}\sin 3-h_{in}-\frac{1}{2}J_{in}^0 \tag{87}$$

Since the unknown functions $h_{in}(\overline{x}_{in},s_{in})$ and $J_n^0(\overline{x}_{in},s_{in})$ are continuous, which can be approximated by NN as

$$h_{in}(\overline{x}_{in},s_{in})=W_{hin}^{*T}S_{hin}(\overline{x}_{in},s_{in})+\varepsilon_{hin}(\overline{x}_{in},s_{in}) \tag{88}$$

$$J_n^0(\overline{x}_{in}, s_{in}) = W_{Jin}^{*T} S_{Jin}(\overline{x}_{in}, s_{in}) + \varepsilon_{Jin}(\overline{x}_{in}, s_{in}) \tag{89}$$

where $W_{hin}^{*T} \in \mathbb{R}^{p_n}$, $W_{Jin}^{*T} \in \mathbb{R}^{q_n}$ are the ideal NN weights, $S_{hin}(x_{in}, s_{in}) \in \mathbb{R}^{p_n}$ and $S_{Jin}(x_{in}, s_{in}) \in \mathbb{R}^{q_n}$ are the basis function vectors, $\varepsilon_{hin}(x_{in}, s_{in}) \in \mathbb{R}$, $\varepsilon_{Jin}(x_{in}, s_{in}) \in \mathbb{R}$ are the bounded approximation errors. Substituting Eqs. (88) and (89) into Eqs. (86) and (87) yields

$$\frac{dJ_{in}^*}{ds_{in}} = 2\gamma_{in}s_{in} + \frac{1}{2\beta_{in}} \sin 3 + 2W_{hin}^{*T} S_{hin} + W_{Jin}^{*T} S_{Jin} + \varepsilon_{in} \tag{90}$$

$$u_i^* = -\gamma_{in}s_{in} - \frac{1}{4\beta_{in}} \sin 3 - W_{hin}^{*T} S_{hin} - \frac{1}{2} W_{Jin}^{*T} S_{Jin} - \frac{1}{2}\varepsilon_{in} \tag{91}$$

where $\varepsilon_{in} = 2\varepsilon_{hin} + \varepsilon_{Jin}$. The adaptive identifier is formulated as

$$\hat{h}_{in}(\overline{x}_{in}, s_{in}) = \hat{W}_{hin}^T S_{hin}(\overline{x}_{in}, s_{in}) \tag{92}$$

where $\hat{h}_{in}(x_{in}, s_{in})$ is the identifier output, $\hat{W}_{hin}^T(t) \in \mathbb{R}^{p_n}$ is the NN weight of identifier.

The weight experiences updates based on the following law:

$$\dot{\hat{W}}_{hin} = \Gamma_{in}(S_{hin}(\overline{x}_{in}, s_{in}) \sin 3 - \sigma_{in} \hat{W}_{hin}) \tag{93}$$

where $\Gamma_{in}$ is a positive-definite constant matrix, $\sigma_{in} > 0$ is constant. The critic is

$$\frac{d\hat{J}_{in}^*(s_{in})}{ds_{in}} = 2\gamma_{in}s_{in} + \frac{1}{2\beta_{in}} \sin 3 + 2\hat{W}_{hin}^T S_{hin}(\overline{x}_{in}, s_{in}) + \hat{W}_{cin}^T S_{Jin}(\overline{x}_{in}, s_{in}) \tag{94}$$

The weight experiences updates based on the following law:

$$\dot{\hat{W}}_{cin} = -\gamma_{cin} S_{Jin}(\overline{x}_{in}, s_{in}) S_{Jin}^T(\overline{x}_{in}, s_{in}) \hat{W}_{cin} \tag{95}$$

where $\gamma_{\text{cin}}$ is a constant. The actor is

$$\hat{u}_i^* = -\gamma_{in}s_{in} - \frac{1}{4\beta_{in}} \sin 3 - \hat{W}_{hin}^T(t) S_{hin}(\overline{x}_{in}, s_{in}) - \frac{1}{2} \hat{W}_{ain}^T S_{Jin}(\overline{x}_{in}, s_{in}) \tag{96}$$

The weight experiences updates based on the following law:

$$\dot{\hat{W}}_{ain} = -S_{Jin}(\overline{x}_{in}, s_{in}) S_{Jin}^T(\overline{x}_{in}, s_{in}) \times (\gamma_{ain}(\hat{W}_{ain} - \hat{W}_{cin}) + \gamma_{cin} \hat{W}_{cin}) \tag{97}$$

These parameters are required to meet the following limitation:

$$\beta_{in} > 0, \gamma_{in} > 4, \gamma_{ain} > \frac{\beta_{in}}{2}, \gamma_{ain} > \gamma_{cin} > \frac{\gamma_{ain}}{2}. \tag{98}$$

Select the Lyapunov function candidate for overall backstepping control as

$$L_{in} = \sum_{j=1}^{n-1} L_{ij} + \frac{1}{4} \sin 4 + \frac{1}{2} \widetilde{W}_{hin}^T \Gamma_{in}^{-1} \widetilde{W}_{hin} + \frac{1}{2} \widetilde{W}_{cin}^T \widetilde{W}_{cin} + \frac{1}{2} \widetilde{W}_{ain}^T \widetilde{W}_{ain} \tag{99}$$

where $\tilde{W}_{hin}(t) = \hat{W}_{hin}(t) - W^*_{hin}$, $\tilde{W}_{cin}(t) = \hat{W}_{cin}(t) - W^*_{Jin}$, $\tilde{W}_{ain}(t) = \hat{W}_{ain}(t) - W^*_{Jin}$.
Compute $\mathfrak{L}$ of $L_{in}$, along with Eqs. (80), (93), (95) and (97), and then apply (96), resulting in the following:

$$\mathcal{L}L_{in} = \sum_{j=1}^{n-1} \mathcal{L}L_{ij} + \sin 3\left(-\gamma_{in}s_{in} - \frac{1}{4\beta_{in}}\sin 3 - \hat{W}_{hin}^T S_{hin} - \frac{1}{2}\hat{W}_{ain}^T S_{Jin} + f_{in} - \mathcal{L}\hat{\alpha}_{n-1}^*\right) +$$
$$\frac{3}{2}s_{in}||\Psi_{in}||^2 + \tilde{W}_{hin}^T(S_{hin}\sin 3 - \sigma_{in}\hat{W}_{hin}) - \gamma_{cin}\tilde{W}_{cin}^T S_{Jin} S_{Jin}^T \hat{W}_{cin} - W_{ain}^T S_{Jin} S_{Jin}^T$$
$$[\gamma_{ain}(\hat{W}_{ain} - \hat{W}_{cin}) + \gamma_{cin}\hat{W}_{cin}] \tag{100}$$

The following expression is derived from Eqs. (100):

$$\mathcal{L}L_{in} \le \sum_{j=1}^{n-1}(-a_{ij}L_{ij} + b_{ij}) - (\gamma_{in} - 4)\sin 4 - \frac{\sigma_{in}}{2\lambda_{\Gamma_{in}^{-1}}^{max}}\tilde{W}_{hin}^T \Gamma_{in}^{-1}\tilde{W}_{hin} - \frac{\gamma_{cin}}{2}\lambda_{S_{Jin}}^{min}\tilde{W}_{cin}^T \tilde{W}_{cin}$$
$$- \frac{\gamma_{cin}}{2}\lambda_{S_{Jin}}^{min}\tilde{W}_{ain}^T \tilde{W}_{ain} + B_{in} \tag{101}$$

where $\lambda_{\Gamma_{in}^{-1}}^{max}$ is the maximal eigenvalue of $\Gamma_{in}^{-1}$, $\lambda_{S_{Jin}}^{min}$ is the minimal eigenvalue of $S_{Jin}S_{Jin}^T$.
And $B_{in} = (\frac{\gamma_{cin}}{2} + \frac{\gamma_{ain}}{2})(W_{Jin}^{*T}S_{Jin})^2 + \frac{\sigma_{in}}{2}||W_{hin}^*||^2 + \frac{1}{4}(\mathcal{L}\hat{\alpha}_{in-1}^*)^4 + \frac{1}{4}\varepsilon_{hin}^4 + \frac{9}{16}$, which satisfied $|B_{in}| \le b_{in}$. Let $a_{in} = \min 4(\gamma_{in} - 4), \sigma_{in}/(\lambda_{\Gamma_{in}^{-1}}^{max}), \gamma_{cin}\lambda_{S_{Jin}}^{min}$, and then Eq. (101) can become the following one

$$\mathcal{L}L_{in} \le \sum_{j=1}^{n}(-a_{ij}L_{ij} + b_{ij}). \tag{102}$$

## STABILITY ANALYSIS

Theorem 1: Consider MASs described by Eq. (13) and subjected to Assumptions 1-2, operating within a directed graph and employing the adaptive laws Eqs. (32), (72) and (97), together with the virtual controllers Eqs. (31) and (71), and the actual controller Eq. (96), the containment control protocol unequivocally guarantees the SGUUB of all signals within the closed-loop system. Furthermore, for a given $\forall t > 0$, tuning the design parameters leads the containment error to converge within an arbitrarily small neighborhood, as expressed:

$$||y_i + \mathbb{L}_1^{-1}\mathbb{L}_2 y_{\ell d}|| \le \bar{\varepsilon} \tag{103}$$

Proof: Consider the overall Lyapunov function L given by:

$$L = \sum_{i=1}^{N}\sum_{j=1}^{n} L_{ij} \tag{104}$$

Define $a_i = \min\{a_{i1}, a_{i2}, \ldots, a_{ij}\}$ and $b_i = \sum_{j=1}^{n} b_{ij}$. Subsequently, Eq. (104) can be expressed as

$$\mathcal{L}L| \le -a_i L + b_i \tag{105}$$

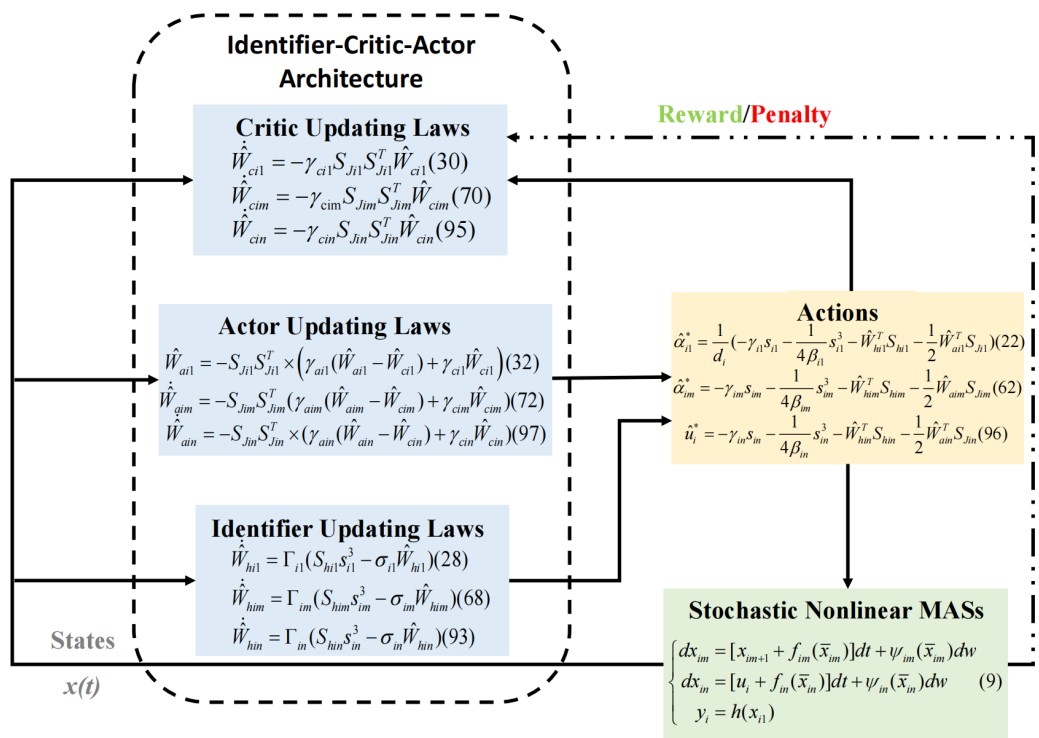

**Figure 2** The RL control scheme.

Based on Lemma 2, the following inequality is deduced from Eq. (105):

$$\mathbb{E}(L) \le e^{-a_i t} L(0) + \frac{b_i}{a_i} \tag{106}$$

$$\mathbb{E}(L) \le E[L(0)] + \frac{b_i}{a_i} \tag{107}$$

For $s_{*1} = [s_{11}, s_{21}, \ldots, s_{N1}]^T$, based on the definition of $L_{in}$ and Eq. (99)

$$\mathbb{E}(||s_{*1}||^4) \le \mathbb{E}(s_{11}^2 + s_{21}^2 + \ldots + s_{N1}^2)^2 \le \mathbb{E}(s_{11}^4 + s_{21}^4 + \ldots + s_{N1}^4) \le 4N(\mathbb{E}[L(0)] + \frac{b_i}{a_i}) \tag{108}$$

where N denotes quantity of follower agents. With Eq. (99), for $\forall \varepsilon > 0$:

$$\mathbb{E}[L(0)] + \frac{b_i}{a_i} \le \frac{\bar{\bar{\varepsilon}}}{8}(\bar{\eta}(\mathbb{L}_1))^4 \tag{109}$$

Taking Eq. (109) and Lemma 3 into account to obtain

$$\mathbb{E}(||y_i + \mathbb{L}_1^{-1}\mathbb{L}_2 y_{\ell d}||) \le \frac{E(||s_{*1}||^4)}{||\bar{\eta}(\mathbb{L}_i)||^4} \le \bar{\varepsilon} \tag{110}$$

The proof is completed and the RL control strategy process diagram is illustrated in Fig. 2.

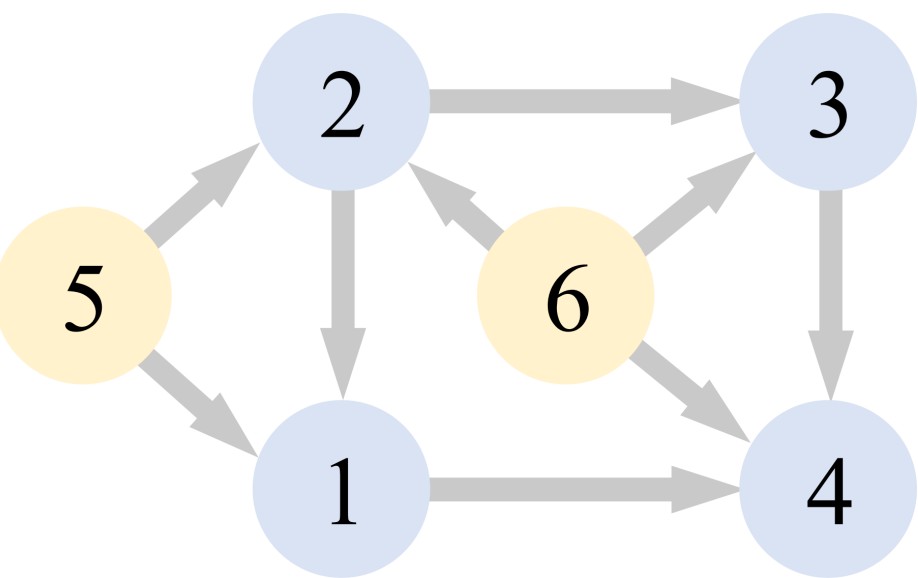

**Figure 3** **Communication graph.**

## SIMULATION EXAMPLE

In this section, the effectiveness of OB, RL and containment control is illustrated by a numerical example. For the nonlinear stochastic MASs consisting of 4 followers and 2 leaders, the following system dynamics are considered:

$$\begin{cases} dx_{i1} = [0.9x_{i2} - 0.8x_{i1}^2 \sin(x_{i2})]dt + \psi_{i1}(\overline{x}_{i1})dw \\ dx_{i2} = [u_i + 0.9\sin(x_{i1})]dt + \psi_{i2}(\overline{x}_{i2})dw \end{cases} \tag{111}$$

where $x_{i1}, x_{i2} \in \mathbb{R}, u \in \mathbb{R}$ is the control input, $\psi_{i1}(\overline{x}_{i1}) = 0.3\sin(x_{i1})$, $\psi_{i2}(\overline{x}_{i2}) = 0.01\sin(0.1\sin(x_{i1}))$. The leaders are defined as:

$$\begin{cases} y_{5r} = 0.1\sin(2t) - 0.1 \\ y_{6r} = 0.45 - 0.5e^{-(t+2)} \end{cases} \tag{112}$$

The communication graph that we used in the simulation is visualized in Fig. 3. According to Fig. 3, the Laplacian matrix as:

$$\mathbb{L} = \begin{bmatrix} 2 & -1 & 0 & 0 & -1 & 0 \\ 0 & 2 & 0 & 0 & -1 & -1 \\ 0 & -1 & 3 & -1 & 0 & -1 \\ -1 & 0 & 0 & 2 & 0 & -1 \\ 0 & 0 & 0 & 0 & 0 & 0 \\ 0 & 0 & 0 & 0 & 0 & 0 \end{bmatrix}. \tag{113}$$

The NN update parameters are designed as: $\gamma_{ai1} = 20, \gamma_{ai2} = 15, \gamma_{ci1} = 14, \gamma_{ci2} = 14, \sigma_{i1} = 14$. The design parameters for the optimized virtual control action $\hat{\alpha}_i^*$ corresponding to Eq. (41) are: $\gamma_{i1} = 12, \beta_{i1} = 5$. The parameters of the optimized actual control action corresponding to Eq. (42) are set as $\gamma_{i2} = 5, \beta_{i2} = 2$.

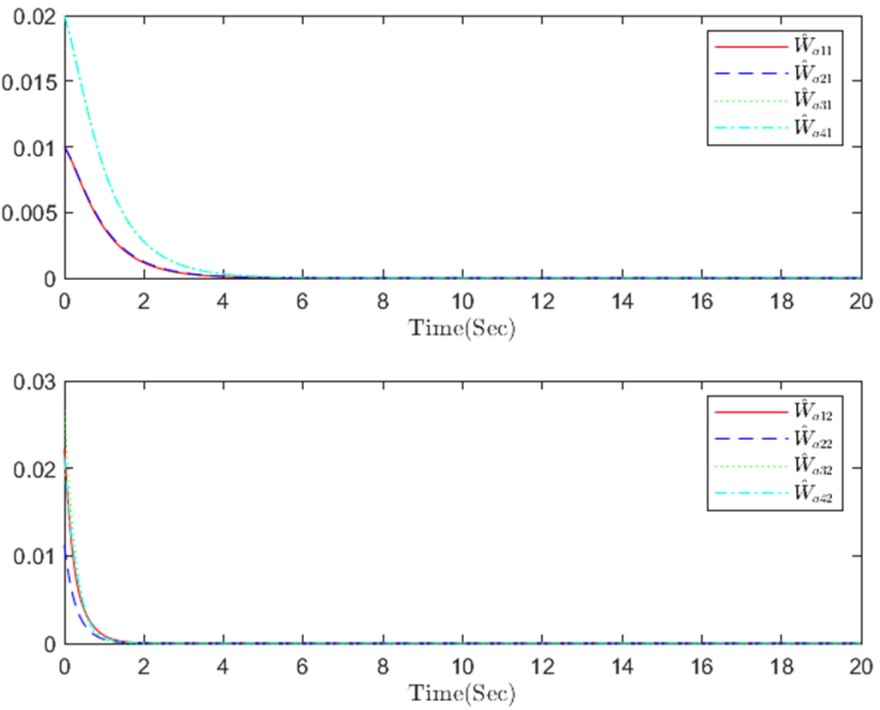

**Figure 4** The actor NN weight a in step m.

The simulation results, illustrating the application of the proposed OB method for stochastic nonlinear MASs, are presented in Figs. 4–12. Figure 4–6 depict the boundedness of the actor, identifier, and critic NN weights. The actor for performing the control action $\hat{\alpha}_i^*$ and the optimized control actor $\hat{u}_i^*$ are illustrated in Figs. 7–8. Figure 9 displays the trajectories of leaders and followers, demonstrating the asymptotic convergence of all followers to the convex hull formed by the leaders. The distributed containment errors are shown in Figs. 10–11. The results verify that all closed-loop system signals are SGUUB. The simulation results demonstrate that the OB method used in MASs can achieve the desired control performance. Besides, Fig. 12 is the error curve without considering the adaptive compensation scheme in this paper. By comparing simulation results, it can be seen that through RL, adjusting the adaptive rate accelerates the convergence speed of the optimization algorithm, allowing sensor errors to converge more quickly.

## CONCLUSION

This article introduces an optimized backstepping control based on RL, which has been developed and applied to a class of nonlinear stochastic strict-feedback MASs experiencing sensor faults. Crafting virtual and actual controls as optimized solutions for their respective subsystems, an overall optimization of the backstepping control has been achieved. To address sensor faults, an adaptive neural network compensation control method has been constructed. Utilizing the RL framework based on neural network approximation, the rules for updating RL have been deduced from the negative gradient of a basic positive

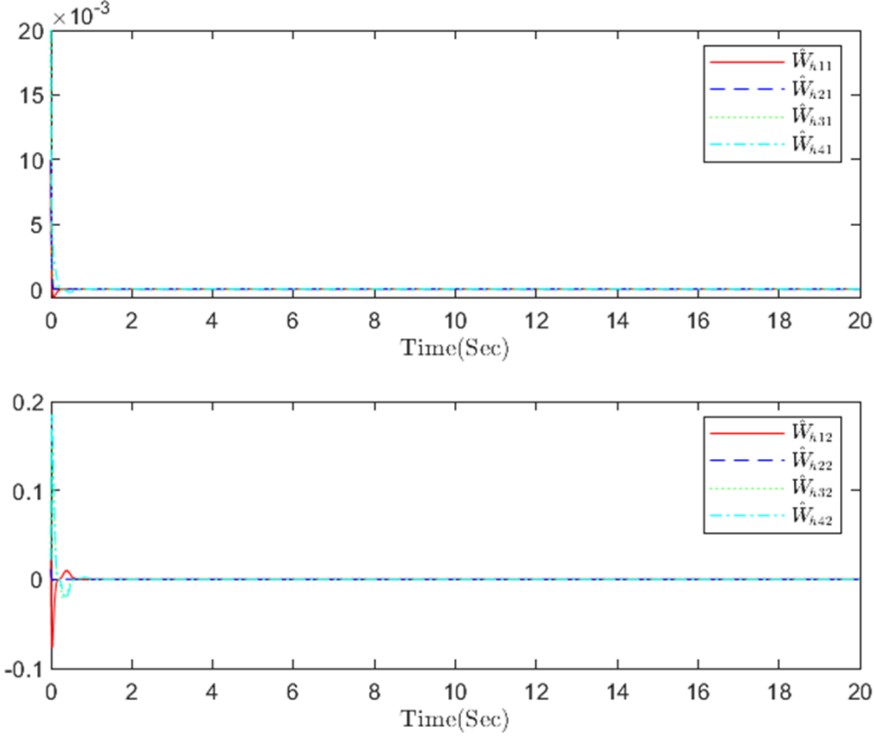

**Figure 5** **The identifier NN weight h in step m.**

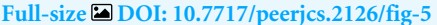

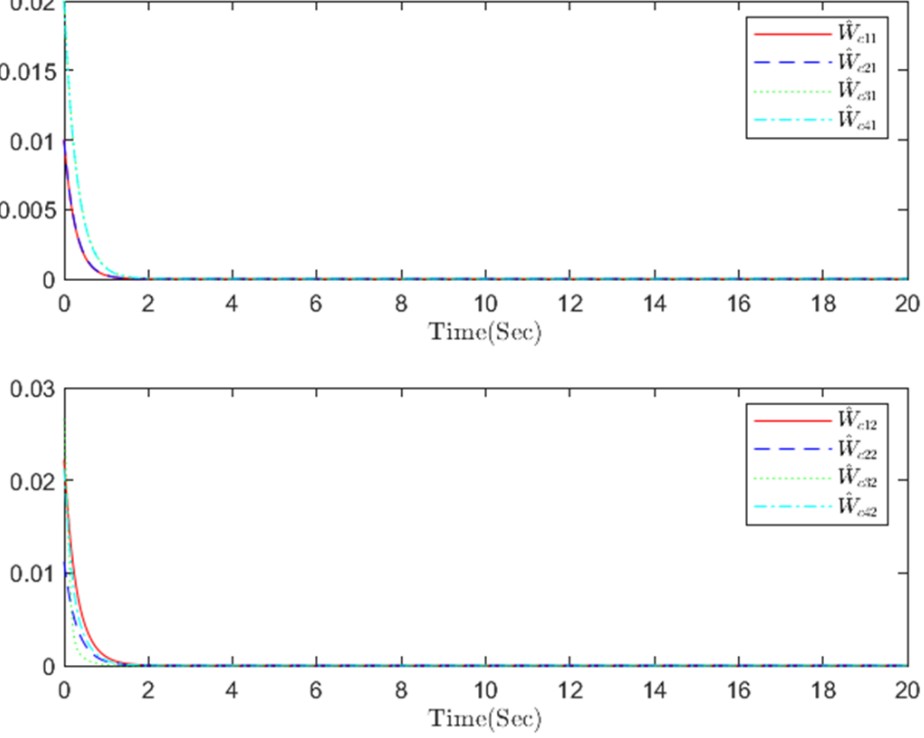

**Figure 6** **The critic NN weight c in step m.**

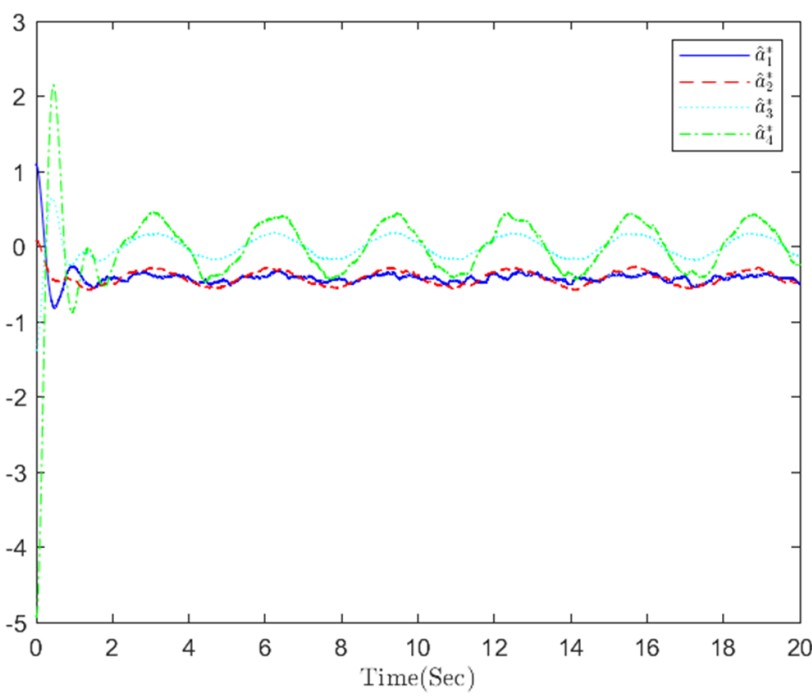

**Figure 7** The optimized virtual control action in step 1.

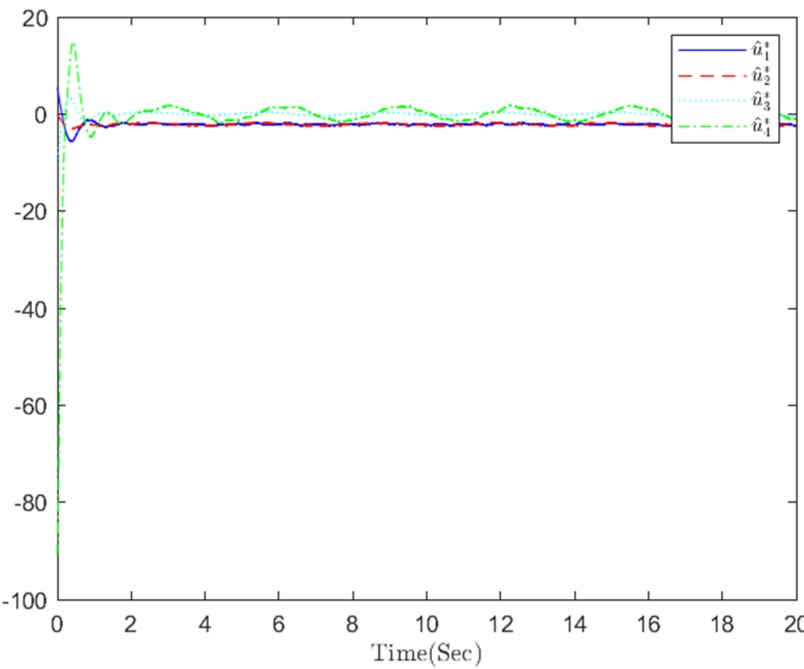

**Figure 8** The optimized actual control action in step 2.

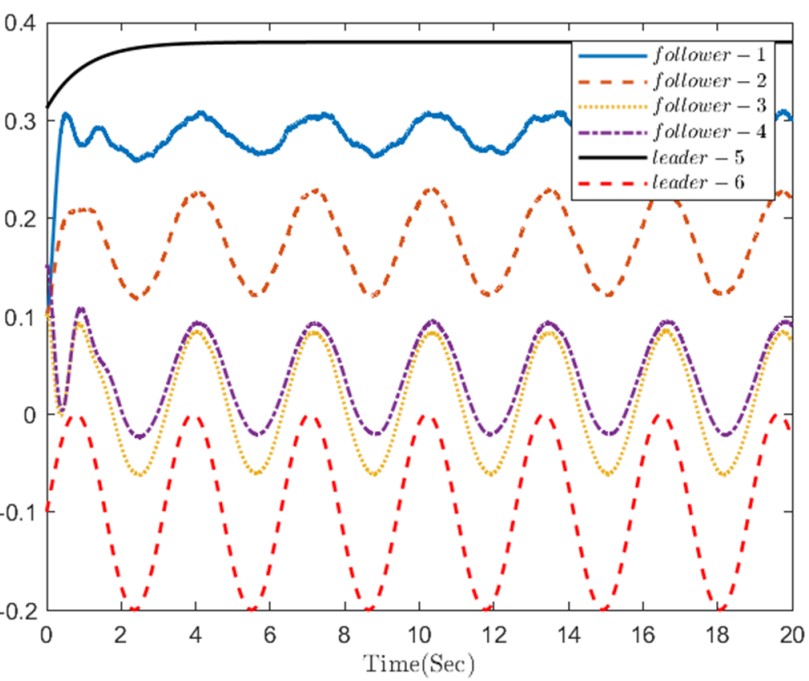

**Figure 9** The trajectories of four followers and two leaders.

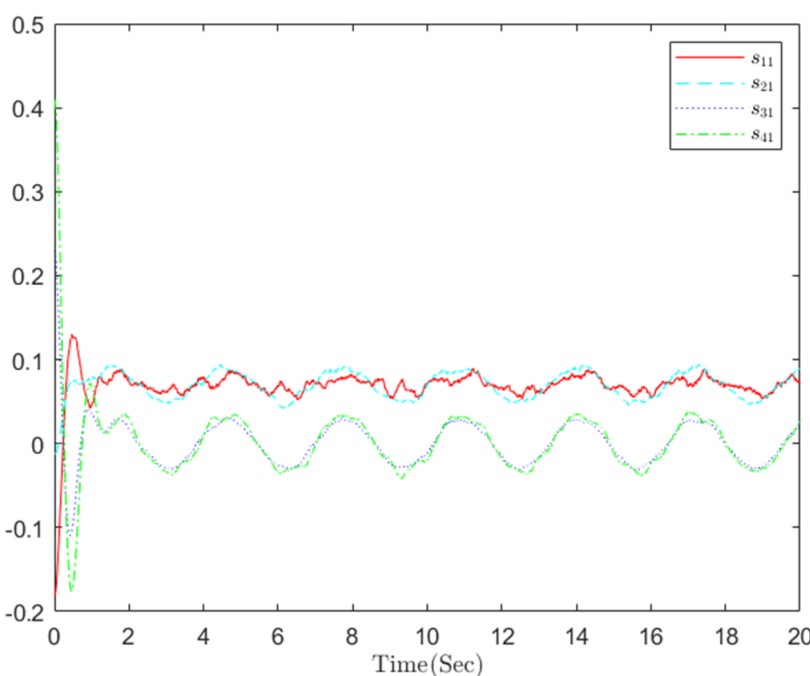

**Figure 10** The distributed containment errors s in step 1.

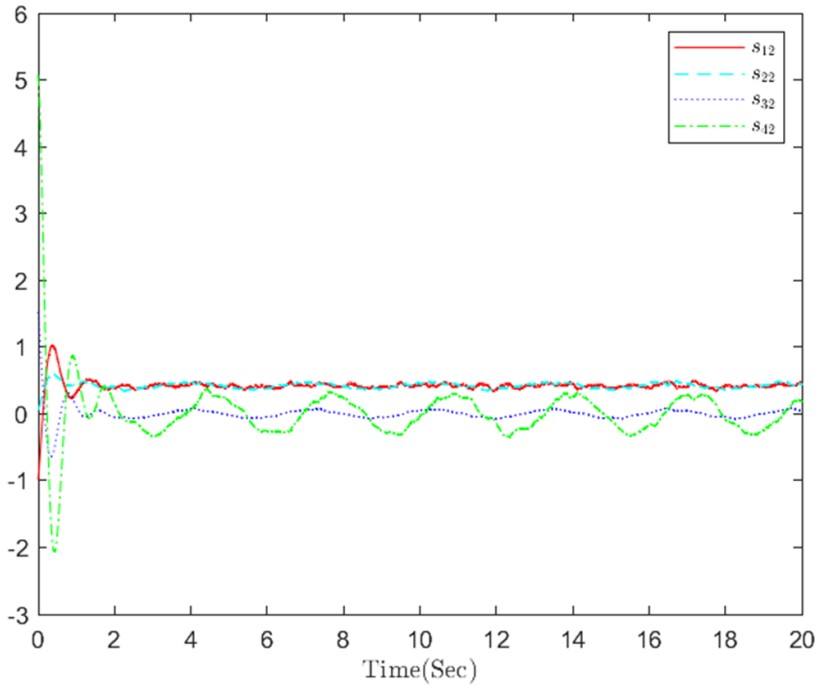

**Figure 11   The distributed containment errors s in step 2.**

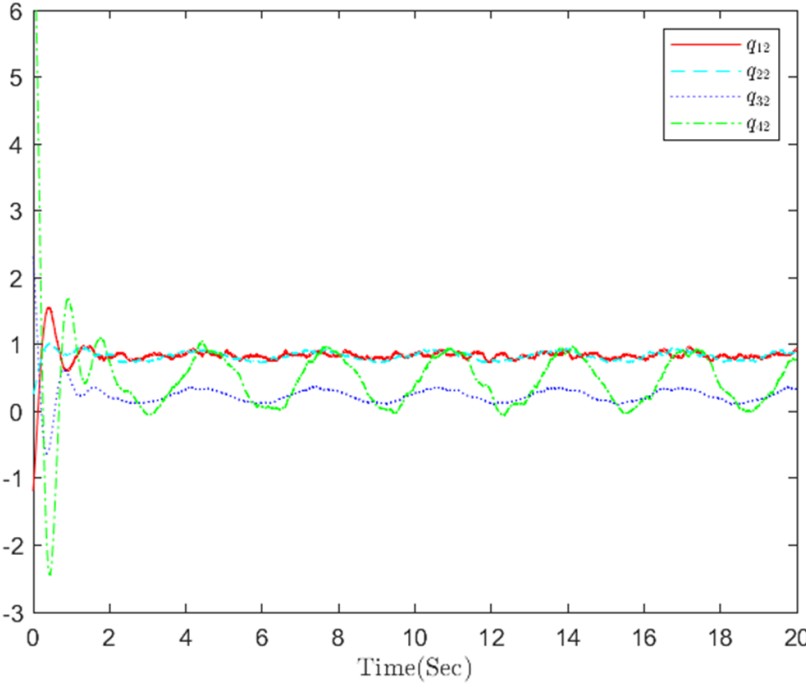

**Figure 12   The distributed containment errors q in step 2.**

function linked to the HJB equation. In comparison with existing methods, not only did this approach significantly simplify the RL algorithm, but it also relaxed the requirements for known dynamics and persistent excitation. Additionally, the proposed control scheme has that the outputs of all followers converge to the dynamic convex hull formed by the leaders.

### Funding
The authors received no funding for this work.

### Competing Interests
The authors declare there are no competing interests.

### Author Contributions
- Guanzong Mo conceived and designed the experiments, performed the experiments, analyzed the data, performed the computation work, prepared figures and/or tables, authored or reviewed drafts of the article, and approved the final draft.
- Yixin Lyu conceived and designed the experiments, performed the experiments, analyzed the data, performed the computation work, prepared figures and/or tables, authored or reviewed drafts of the article, and approved the final draft.

### Data Availability
  The relevant code are available in Supplementary File.

### Supplemental Information
Supplemental information for this article can be found online at http://dx.doi.org/10.7717/peerj-cs.2126#supplemental-information.

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
