# Peer review of "Adaptive resilient containment control using reinforcement learning for nonlinear stochastic multi-agent systems under sensor faults"

_PeerJ Computer Science, doi:10.7717/peerj-cs.2126_

## Round 0.1 · original submission · Major Revisions

According to the reviewers comments, the manuscript must be revised.

Reviewer 1 ·

Basic reporting

The manuscript entitled “Adaptive resilient containment control using reinforcement learning for nonlinear stochastic multi-agent systems under sensor faults” has been investigated in detail. The paper proposes an optimized backstepping control strategy for nonlinear stochastic strict-feedback multi-agent systems (MASs) with sensor faults. It incorporates adaptive neural network compensation control and reinforcement learning (RL) based on neural network approximation. Theoretical analysis based on stochastic Lyapunov theory is claimed to demonstrate the effectiveness of the proposed strategy. There are some points that need further clarification and improvement:
1) The paper lacks a clear and compelling rationale for addressing the proposed problem of controlling nonlinear stochastic strict-feedback multi-agent systems with sensor faults. It fails to establish the significance of this problem in the context of existing research or real-world applications.
2) The methodology is inadequately described, making it difficult to understand the proposed control strategy comprehensively. Key details regarding the design of virtual and actual controls, adaptive neural network compensation, and reinforcement learning framework are either missing or poorly explained.
3) The utilization of reinforcement learning (RL) based on neural network approximation lacks justification, especially regarding its suitability for addressing sensor faults in MASs. The connection between RL update rules and the Hamilton-Jacobi-Bellman (HJB) equation is not adequately elaborated.
4) While the paper claims theoretical analysis based on stochastic Lyapunov theory, the details provided are insufficient to assess the validity and rigor of the analysis. Important aspects such as stability proofs and convergence properties are not adequately demonstrated or explained.

Experimental design

The numerical simulations presented to validate the proposed control strategy are not thoroughly discussed or analyzed. Insufficient detail is provided regarding the simulation setup, performance metrics, and comparison with existing methods, limiting the assessment of the proposed approach's effectiveness.

The paper suffers from a lack of clarity, coherence, and rigor in presenting the proposed control strategy for nonlinear stochastic strict-feedback multi-agent systems with sensor faults.

Validity of the findings

“Discussion” section should be added in a more highlighting, argumentative way. The author should analysis the reason why the tested results is achieved.

This study may be proposed for publication if it is addressed in the specified problems.

Additional comments

The manuscript entitled “Adaptive resilient containment control using reinforcement learning for nonlinear stochastic multi-agent systems under sensor faults” has been investigated in detail. The paper proposes an optimized backstepping control strategy for nonlinear stochastic strict-feedback multi-agent systems (MASs) with sensor faults. It incorporates adaptive neural network compensation control and reinforcement learning (RL) based on neural network approximation. Theoretical analysis based on stochastic Lyapunov theory is claimed to demonstrate the effectiveness of the proposed strategy. There are some points that need further clarification and improvement:
1) The paper lacks a clear and compelling rationale for addressing the proposed problem of controlling nonlinear stochastic strict-feedback multi-agent systems with sensor faults. It fails to establish the significance of this problem in the context of existing research or real-world applications.
2) The methodology is inadequately described, making it difficult to understand the proposed control strategy comprehensively. Key details regarding the design of virtual and actual controls, adaptive neural network compensation, and reinforcement learning framework are either missing or poorly explained.
3) The utilization of reinforcement learning (RL) based on neural network approximation lacks justification, especially regarding its suitability for addressing sensor faults in MASs. The connection between RL update rules and the Hamilton-Jacobi-Bellman (HJB) equation is not adequately elaborated.
4) While the paper claims theoretical analysis based on stochastic Lyapunov theory, the details provided are insufficient to assess the validity and rigor of the analysis. Important aspects such as stability proofs and convergence properties are not adequately demonstrated or explained.
5) The numerical simulations presented to validate the proposed control strategy are not thoroughly discussed or analyzed. Insufficient detail is provided regarding the simulation setup, performance metrics, and comparison with existing methods, limiting the assessment of the proposed approach's effectiveness.
6) The paper suffers from a lack of clarity, coherence, and rigor in presenting the proposed control strategy for nonlinear stochastic strict-feedback multi-agent systems with sensor faults.
7) “Discussion” section should be added in a more highlighting, argumentative way. The author should analysis the reason why the tested results is achieved.
This study may be proposed for publication if it is addressed in the specified problems.

Reviewer 2 ·

Basic reporting

no comment

Experimental design

no comment

Validity of the findings

no comment

Additional comments

The paper "Adaptive resilient containment control using reinforcement learning for nonlinear stochastic multi-agent systems under sensor faults" proposes an optimized feedback control strategy developed for a category of strict feedback nonlinear stochastic multi-agent systems (MAS) with sensor faults. An adaptive neural network (NN) control method is considered for sensor troubleshooting. A reinforcement learning (RL) framework based on a neural network approximation is used. A theoretical analysis based on stochastic Lyapunov theory demonstrates the semi-global uniform ultimate boundedness (SGUUB) of all signals in a closed system and illustrates the convergence of all follower outputs to the dynamic convex hull defined by the leaders. The effectiveness of the proposed control strategy is verified by numerical simulation. In general, the article meets the requirements of the journal and can be recommended for publication in its current form.

Reviewer 3 ·

Basic reporting

This paper proposes an optimized backstepping control strategy designed for a category of
nonlinear stochastic strict-feedback multi-agent systems (MASs) with sensor faults. The
plan formulates optimized solutions for the respective subsystems by designing both
virtual and actual controls, achieving overall optimization of the backstepping control. To
address sensor faults, an adaptive neural network (NN) compensation control method is
considered. The reinforcement learning (RL) framework based on neural network
approximation is employed, deriving RL update rules from the negative gradient of a
simple positive function correlated with the Hamilton-Jacobi-Bellman (HJB) equation. This
signiûcantly simpliûes the RL algorithm while relaxing the constraints for known dynamics
and persistent excitation. The theoretical analysis, based on stochastic Lyapunov theory,
demonstrates the semi-global uniform ultimate boundedness (SGUUB) of all signals within
the enclosed system, and illustrates the convergence of all follower outputs to the dynamic
convex hull deûned by the leaders. I have some major concerns in the manuscript:

The paper appears to be rooted in mathematical proofs and is aligned with the mathematical domain. However, to connect it with computing and information systems, several key aspects need to be addressed. While reinforcement learning (RL) is mentioned, there is a lack of description of the RL framework, neural network design, simulation setup, parameters, and designs.

Firstly, it's essential to understand how RL works in this context. Does the paper create a custom environment for the RL algorithm to operate in? Additionally, what are the different libraries that the authors used in the simulation? These details are crucial for bridging the gap between the mathematical domain and computing.

Moreover, while there is a high-level description of the proposed algorithm, the authors should provide a complete alignment with computing criteria. This includes detailing the computational aspects of the algorithm, such as implementation details, computational complexity analysis, and optimization techniques used.

Furthermore, the results section lacks a clear description of improvements in containment control. Without graphs or tables showcasing the results, it becomes challenging for the reader to understand the superiority of this paper. Including visual representations of the results and comparing them with recent benchmarks would greatly enhance the clarity and impact of the findings.

In order to address these issues, a dedicated section regarding data, environment, simulations settings, neural network descriptions, and the use of specific RL algorithms (e.g., DQN, DDPG, PPO) is necessary. This section should provide detailed insights into how the algorithm operates within the computing and information systems context.

Overall, enhancing the computational aspects, providing clear results with graphical representations, and detailing the RL framework and simulation setup will strengthen the paper's alignment with computing and information systems and improve its clarity and impact.

Experimental design

See basic reporting.

Validity of the findings

See basic reporting.

---

## Round 0.2 · accepted · Accept

Based on the reviewers comments, the manuscript can be accepted.

Reviewer 1 ·

Basic reporting

My comments have been addressed. It is acceptable in the present form.

Experimental design

My comments have been addressed. It is acceptable in the present form.

Validity of the findings

My comments have been addressed. It is acceptable in the present form.

Reviewer 3 ·

Basic reporting

The authors have carefully revised the manuscript. I am happy to suggest the acceptance of the article.

Experimental design

No comment

Validity of the findings

No comment